# Simian Foamy Virus Prevalence and Evolutionary Relationships in Two Free-Living Lion Tamarin Populations from Rio de Janeiro, Brazil

**DOI:** 10.3390/v17081072

**Published:** 2025-07-31

**Authors:** Déa Luiza Girardi, Thamiris Santos Miranda, Matheus Augusto Calvano Cosentino, Caroline Carvalho de Sá, Talitha Mayumi Francisco, Bianca Cardozo Afonso, Flávio Landim Soffiati, Suelen Sanches Ferreira, Silvia Bahadian Moreira, Alcides Pissinatti, Carlos Ramon Ruiz-Miranda, Valéria Romano, Marcelo Alves Soares, Mirela D’arc, André Felipe Santos

**Affiliations:** 1Departamento de Genética, Universidade Federal do Rio de Janeiro, Rio de Janeiro 21941-617, RJ, Brazil; dluiza.gii@gmail.com (D.L.G.); thamirismiranda02@gmail.com (T.S.M.); macosen@gmail.com (M.A.C.C.); 2Programa de Genética e Virologia Tumoral, Instituto Nacional de Câncer (INCA), Rio de Janeiro 0230-130, RJ, Brazil; caroline.sa@ensino.inca.gov.br (C.C.d.S.); masoares@inca.gov.br (M.A.S.); 3Associação Mico-Leão-Dourado, Silva Jardim, Rio de Janeiro 28820-000, RJ, Brazil; talithamayumi@gmail.com (T.M.F.); flaviosoffiati@gmail.com (F.L.S.); suelen.sferreira@gmail.com (S.S.F.); cruiz@uenf.br (C.R.R.-M.); 4Campos dos Goytacazes, Laboratório de Ciências Ambientais, Centro de Biociências e Biotecnologia, Universidade Estadual do Norte Fluminense Darcy Ribeiro-UENF, Rio de Janeiro 28013-602, RJ, Brazil; biavetufes@gmail.com; 5Centro de Primatologia do Rio de Janeiro, Instituto Estadual do Ambiente, Guapimirim 25948-395, RJ, Brazil; silviabm.inea@gmail.com (S.B.M.); alcidespissinatti@gmail.com (A.P.); 6IMBE, Aix Marseille University, Avignon University, CNRS, IRD, 13397 Marseille, France; valeria.romano@imbe.fr

**Keywords:** spumavirus, wild primates, South America, prevalence, retroviruses

## Abstract

Simian foamy virus (SFV) is a retrovirus that infects primates. However, epidemiological studies of SFV are often limited to captive populations. The southeastern Brazilian Atlantic Forest is home to both an endemic, endangered species, *Leontopithecus rosalia*, and an introduced species, *Leontopithecus chrysomelas*, to which no data on SFV exist. In this study, we assessed the molecular prevalence of SFV, their viral load, and their phylogenetic relationship in these two species of primates. Genomic DNA was extracted from 48 oral swab samples of *L. chrysomelas* and 102 of *L. rosalia*. Quantitative PCR (qPCR) was performed to diagnose SFV infection and quantify viral load. SFV prevalence was found to be 23% in *L. chrysomelas* and 33% in *L. rosalia*. No age-related differences in prevalence were observed; however, *L. rosalia* showed a higher mean viral load (3.27 log10/10^6^ cells) compared to *L. chrysomelas* (3.03 log10/10^6^ cells). The polymerase gene sequence (213 pb) of *L. rosalia* (SFVlro) was clustered within a distinct SFV lineage found in *L. chrysomelas.* The estimated origin of SFVlro dated back approximately 0.0836 million years ago. Our study provides the first molecular prevalence data for SFV in free-living *Leontopithecus* populations while offering insights into the complex evolutionary history of SFV in American primates.

## 1. Introduction

Simian foamy virus (SFV) is a retrovirus classified within the *Simiispumavirus* genus under the subfamily *Spumaretrovirinae* [1,2]. SFV was first described in 1954 [3] and isolated in 1955 [4], and, since then, numerous non-human primates have been described as hosts, including prosimians, Afro-Eurasian primates (formerly known as Old World primates) [5,6,7], and American primates (AP, formerly known as neotropical primates) [8,9,10,11]. AP are highly diverse, with approximately 187 species distributed within 23 genera under five families (Aotidae, Atelidae, Callitrichidae, Cebidae, and Pitheciidae), according to molecular analyses [12,13]. Among AP, the first SFV was detected in 1973, in a culture of brain cells from *Ateles* sp. [14]. Nevertheless, it was only after 34 years that the complete genome of this virus was obtained [15,16,17]. A total of 41 species of AP (~22% of the total number of AP) have molecular evidence of SFV infection; however only five of them have complete viral genomes sequenced [10]. The available sequences include, in addition to the one that infects *Ateles* sp. (SFVasp) [10,17], the SFV infecting *Callithrix jacchus* (SFVcja) [18], *Sapajus xanthosternos* (SFVsxa) [15], *Brachyteles arachnoides* (SFVbar) [16], and *Saimiri sciureus* (SFVssc) [18].

In terms of prevalence in AP, still little is known. It is estimated that the mean prevalence of SFV in captive AP ranges from 23 to 61% [8,11,14,19,20], and among the free-living AP, it ranges from 16 to 29% [8,9,20]. The scarce studies on natural SFV infections are a limiting factor in our understanding of the virus’ epidemiology [10]. Furthermore, although only a few complete and partial SFV genomes from AP are currently available in literature, phylogenetic analyses suggest that SFV generally follows a co-speciation model, as observed in Afro-Eurasian primates [7,9,11,20]. AP arrived in the Americas approximately 40 million years ago [21], diverging between 41.1–22.7 million years ago [21]. Due to the relatively recent speciation of AP and the ecological overlap among extant species—characterized by shared habitats—there is evidence of cross-species transmission events of SFV between AP species and genera [7,20,22,23].

Over 40% of AP species are endangered, including lion tamarins (genus *Leontopithecus*) [24]. The *Leontopithecus* genus is composed of four species: *L. rosalia* (golden lion tamarins), *L. chrysomelas* (golden-faced lion tamarins), *L. chrysopygus* (black lion tamarins), and *L. caissara* (black-faced lion tamarins) [25]. *L. rosalia* is endemic to the Atlantic Forest in Rio de Janeiro, Brazil [26]. They are arboreal and territorial primates, living in small familial groups. Classified as endangered by the International Union for Conservation of Nature in 2022 [13], the non-governmental organization Associação Mico-Leão-Dourado (AMLD) has been working since 1992 on the conservation of this species in Rio de Janeiro [27]. The *L. rosalia* population was considered almost extinct in 1960, a situation that led to the creation of the first biological reserve in Brazil, Poço das Antas. Since then, long-term research has been implemented to understand the behaviors of these primates [26]. In 1983, the reintroduction of captive-born primates began, and in 2003, the status of *L. rosalia* changed from critically endangered to endangered [28]. Nowadays, 19 groups of *L. rosalia* are monitored by AMLD in order to understand where and how they live [29].

*L. chrysomelas* also inhabit the Atlantic Forest, being native to the state of Bahia, Brazil, and are considered endangered in this location. Despite that, some *L. chrysomelas* have been found in the city of Niterói, Rio de Janeiro, Brazil, as a result of an introduction by a collector in the mid-1990s [28]. These primates have been established in fragments of the Atlantic Forest in this city, being considered an introduced species. In Niterói, free-living *L. chrysomelas* are monitored, captured, and transported to centers of preservation in their natural habitat in Bahia by institutions such as Centro de Primatologia do Rio de Janeiro (CPRJ), Fundação Pri-Matas, Instituto Chico Mendes de Conservação da Biodiversidade (ICMBio), and Instituto Estadual do Ambiente (Inea) [27]. *Leontopithecus* primates usually feed from fruits, insects, and small vertebrates and may occasionally consume bird eggs [30]. *L. rosalia* usually live in groups composed of a breeding pair and their offspring, and their family groups are composed of seven individuals on average [26]. They are arboreal and can be found between 3 to 10 m above the ground and tend to sleep in tree holes abandoned by other species [30]. *L. chrysomelas* form groups of 4 to 8 individuals [31]. They are also arboreal and select tall forests for sleeping at night [32]. Both species have ecological importance and are threatened by deforestation, habitat fragmentation, illegal trafficking, and diseases. They also face threats in competition for territory and resources or exposure to pathogens due to interactions with other primate species, like those from the *Callithrix* genus [33]. Although there are studies demonstrating the infection and prevalence of SFV in lion tamarins, they are limited to a few individuals and/or captive animals [8,11,20,34]. The objective of this study is to describe new data on the prevalence of SFV in free-living *L. rosalia* and *L. chrysomelas*, as well their viral load and their phylogenetic relationship.

## 2. Materials and Methods

### 2.1. Sample Collection

We collected oral swabs from 102 individuals of *L. rosalia* (48 females and 54 males) in Silva Jardim and from 48 individuals of *L. chrysomelas* (19 females and 29 males) in Niterói, both cities in the state of Rio de Janeiro, Brazil. Captures of *L. rosalia* and *L. chrysomelas* were organized and managed by specialists of Associação Mico-Leão-Dourado (AMLD) and by the Centro de Primatologia do Rio de Janeiro (CPRJ), respectively, which have years of fieldwork experience. Sample collection occurred between February and September 2021 from animals that were habituated to human presence. Animals were captured individually with Tomahawk^®^ traps with banana baits, as described in [35]. All captured tamarins from Silva Jardim/RJ were taken to the AMLD field laboratory for routine veterinary examinations before release. Free-living *L. chrysomelas*, after capture, were placed in appropriate enclosures until their transportation to centers of preservation in their natural habitat in Bahia at CPRJ. The collection was made immediately after the capture. Animals were anesthetized with an injection of ketamine (10–15 mg/kg) in the caudal region, and general information was collected for all sampled animals, such as identification number, group, species, age, sex, weight, and clinical conditions. While the animals were still anesthetized, samples of oral swabs were collected. Animals aged four to nine months were considered juveniles, nine to twelve months as subadults, and aged >twelve months as adults. For estimating age from animals whose birth dates were unknown, measurements such as body size and weight, identification pattern, and postnatal ossification were also used [36,37,38]. The health status of the animals was obtained through observations made by the veterinarians. They mainly observed respiratory conditions, animal weight, and the presence of dysbiosis.

This project was approved by the Ethics Committee on the Use of Animals (CEUA) of UFRJ (reference number 037/14). All procedures were conducted in full compliance with federal permits issued by the Brazilian Ministry of the Environment (SISBIO 75941-4), and samples were collected following the national guidelines and provisions of IBAMA (Instituto Brasileiro do Meio Ambiente e dos Recursos Naturais Renováveis, Brazil; permanent license number 11375–1). We did not provide the geographic coordinates of individuals or groups for safety reasons.

### 2.2. Sample Processing and Analysis of Genomic DNA Integrity

Collected swabs were placed in 1.5 mL microcentrifuge tubes (Kasvi, Curitiba, PR, Brazil) containing 500 µL of RNAlater (Invitrogen, Thermo Fisher Scientific, Waltham, MA, USA). All samples were stored at room temperature and sent to the Laboratory of Diversity and Viral Diseases (LDDV) at the Universidade Federal do Rio de Janeiro (UFRJ), Rio de Janeiro, Brazil, to be stored at −80 °C until processing. Genomic DNA (gDNA) was extracted from oral swabs using the PureLink^®^ Genomic DNA kit (ThermoFisher Scientific, Grand Island, NY, USA) according to the manufacturer’s specifications. After extraction, samples had their contaminants (PCR inhibitors) removed with the PureLink™ PCR Purification Kit (Invitrogen). Samples were then quantified using Nanodrop ND-1000 (ThermoFisher Scientific) and stored at −20 °C. gDNA integrity for PCR analysis was checked by PCR amplification of the mitochondrial constitutive gene cytochrome B (cytB), as previously described [8]. *cytB*-positive samples were considered suitable for performing quantitative real-time PCR (qPCR) diagnostic for AP SFV.

### 2.3. Diagnostic qPCR

As described by Muniz and collaborators [39], a qPCR assay was used to simultaneously detect and quantify AP SFV DNA of the oral mucosa epithelium cells, obtained through oral swabs. TaqMan (Thermo Fisher Scientific) was used with primers and a probe for the amplification of a 124 bp fragment located in the virus polymerase (*pol*) gene (integrase region) [39]. The construction of the standard curve was performed through serial dilutions of 10^8^ down to 10^1^ copies of a pCR4-TOPO plasmid containing an insert corresponding to a 6400 bp genomic fragment of the Saimiri SFV (SFVssc) containing the *pol* gene. The plasmid was purified with the QIAGEN™ Plasmid Midi Kit (QIAGEN, Hilden, Germany) according to the manufacturer’s protocol.

All reactions were set up in a total volume of 21 μL containing 10 μL of GoTaq^®^ Probe qPCR Master Mix 20X (Promega, São Paulo, Brazil), 1 μL of primers and probe, and 10 μL of gDNA. Reactions were carried out in 96-well optical microtiter plates on a 7500 Real-Time PCR platform (Applied Biosystems, Waltham, MA, USA), following uniform cycling parameters previously described [34]. The sensitivity of the assay was 100 copies of SFV/reaction, as determined previously [34]. Samples with more than 100 viral copies per 10^6^ cells were considered positive for SFV infection in the qPCR. To calculate the integrated viral load in 10^6^ cells, the amount of DNA was converted from nanograms to picograms (pg) and multiplied by 1 million cells, with a cell having a mean of 6 pg of DNA [40].

### 2.4. PCR of the Pol Region

A nested PCR was performed to amplify a larger fragment of SFV *pol* in *Leontopithecus* for subsequent sequencing and phylogenetic analysis. The first PCR round aimed to amplify a 486 bp fragment of the *pol* gene, and the second round amplified a 404 bp fragment internal to the first round. The primers used were specific for SFV *pol* infecting the Cebidae and Callitrichidae families, and conditions were the same as those previously reported for other generic PCRs for SFV *pol* [11], as can be seen in the table below (Table 1). Successfully amplified fragments were purified with the E-Gel CloneWellTM (Invitrogen) according to manufacturer instructions.

Purified samples were submitted to sequencing using the Big Dye Terminator Cycle Sequencing kit v3.1 (Life Technologies, Van Allen Way, Carlsbad, CA, USA). The plate was precipitated and stored at −20 °C, protected from light until sequencing in an 3130XL genetic analyzer (Life Technologies, Foster City, CA, USA). The obtained sequences were assembled and edited in SeqMan 7.0 (DNASTAR, Madison, WI, USA).

### 2.5. Phylogenetic and Timescale Analyses

The assembled sequences were analyzed with the Basic Local Alignment Search Tool (BLAST) against the complete GenBank dataset available as of 1 September 2024. Up to 100 SFV *pol* sequences were retrieved, and, with the removal of identical sequences, a dataset of 90 sequences was built for subsequent evolutionary analyses. The SFV *pol* sequence of *Prosimiispumavirus otocrafo* (Brown greater galago prosimian foamy virus, accession number: NC_039023.1) was added to this dataset to root the phylogeny. Sequences were aligned with sequences representing different SFV strains, comprising a comprehensive dataset of 92 *pol* sequences with 213 nucleotide positions (Appendix A).

Sequences were aligned with MAFFT v7.505 [41] and subsequently trimmed with TrimAl v.1.4 [42] with a gap threshold of 0.9, removing sites from alignments composed of gaps in 10% or more sequences. The multiple sequences alignment had its phylogenetical signal evaluated by the following analyses: likelihood mapping plots with a random sample of 1000 in IQ-Tree v.2.1.4 [43], alignment confidence inferred by the Guidance 2 webserver [44], and verification of substitution saturation with DAMBE v.7.3.32 [45]. Maximum likelihood trees were inferred with IQ-Tree v.2.1.4 [46], with the best fit model suggested by ModelFinder [47]. Node supports were estimated using the SH-type approximate likelihood ratio test (SH-aLRT) [48] and ultrafast bootstrap [49] with 10,000 replicates.

As the phylogenetic pattern for the SFV observed at the SFVlro clade was not consistent with the ancient within-species diversity model expected, we inferred a time-scaled phylogenetic tree using RelTime-ML [50,51,52] to obtain chronological information. This analysis was performed in MEGA11, and estimated host divergence dates were used to calibrate internal nodes of the viral tree, following [21]. The calibration dates were chosen considering the host evolutionary history, in points found in our SFV phylogeny, which mirrors the primates’ major splits found by Kuderna [21].

A Jukes–Cantor substitution model with a Gamma distribution, invariable rate variation model (JC+G4+I), and six calibration constraints were used to build the time tree. The ancestral node of the *Macaca* SFV was calibrated with a normal distribution centered at 5.49 million years ago (Mya) and a standard deviation of 0.35 (95% CI 4.9 to 6.08). Similarly, the ancestral clade of the *Pongo* SFV was calibrated with a normal distribution of mean 1.55 Mya and a standard deviation of 0.15 (95% CI 1.26 to 1.84). For the *Pan* SFV ancestral node, a calibration of 8.01 Mya with a standard deviation of 0.45 (95% CI 7.13 to 8.89) was applied. The shared ancestral node of the *Pan* and *Pongo* SFVs was calibrated at 20.32 Mya with a standard deviation of 0.85 (95% CI 18.65 to 21.99), while the shared ancestral node of the Cercopithecidae, Hylobatidae, and Hominidae SFVs was set to 31.08 Mya, with a standard deviation of 0.95 (95% CI 29.12 to 33.04). Finally, the ancestral node of Platyrrhini and Catarrhini SFVs was calibrated at 39.03 Mya, with a standard deviation of 1.1 (95% CI 36.87 to 41.19). All calibration points and standard deviations were calculated following [21].

The phylogenetic tree and time tree were visualized using ggtree v.3.10 [53] in R v.4.3.2. The original nucleotide dataset, final alignment file, generated tree files, and timescale files are provided in Appendix A.

### 2.6. Statistical Analyses

Similar to Ref. [54], we applied linear mixed-effect models to test whether morphological variables (age, sex, and body weight) correlated with proviral load, including species as a random effect to control for pseudoreplication. The model was performed using the “lmer” function in the “lme4” package [55] in R. To evaluate the validity of the model, we performed a series of diagnostic tests, including assessments of residual homoscedasticity, normality of the data distribution, and variance inflation factors (VIF), which were consistently near or below 1. The diagnostic results confirmed that the data followed a normal distribution, and no influential observations were detected.

## 3. Results

### 3.1. Study Population

Oral swab samples were collected from 150 *Leontopithecus* specimens, 48 of which were *L. chrysomelas* housed at CPRJ, and 102 were *L. rosalia* captured at AMLD. While for *L. rosalia,* the distribution of samples was similar between sexes (48 females and 54 males), for *L. chrysomelas,* there was a predominance of samples from male animals (*n* = 29, 60%). The age categories were obtained for both species. In *L. rosalia*, a predominance of adults was observed (*n* = 48, 47%), as well as morphological information and field data, all described in Table 2. For the *L. chrysomelas* population, there was also a predominance of adults (*n* = 42, 87%).

The *L. rosalia* were in good physical condition, with a median weight of 521 g, and no changes were observed in the physical examination. *L.* chrysomelas were also in good condition and had a median weight of 603 g. All information is summarized in Table 2.

### 3.2. SFV Prevalence

All samples successfully underwent PCR amplification of the constitutive cytB gene and were deemed suitable for SFV diagnosis via qPCR. The overall prevalence of SFV infection in the *Leontopithecus* genus was 30% (45 cases of 150 total individuals). In *L. chrysomelas*, the overall prevalence of SFV was 23% (11/48, Figure 1A), with 31% of females (6/19) and 17% of males (5/29) testing positive (Figure 1B). The overall SFV prevalence of *L. rosalia* was 33% (34/102) (Figure 1A), with both females (16/48) and males (18/54) showing the same prevalence of 33% (Figure 1B). There was no statistical difference between SFV prevalence in *L. rosalia* and *L. chrysomelas* (X^2^ = 1.23, *p* = 0.27).

In both species, prevalence varied by age category. In *L. chrysomelas*, juveniles exhibited a 50% prevalence, though only two individuals were sampled, with a female infected and a male uninfected. Subadults also had a 50% prevalence, based on a sample of four individuals, with no females (*n* = 2) infected and the males (*n* = 2) testing positive. Adult females had a prevalence of 28% (5/18), while adults males had a prevalence of 8% (2/24). In the *L. rosalia* population, juveniles had a prevalence of 29% (9/31), subadults had 22% (5/23), and adults showed a prevalence of 42% (20/48) (Figure 1C). There was no signifincant difference in SFV prevalence between the age categories (X^2^ = 3.15, *p* = 0.21). When comparing the prevalence between females and males within each category for *L. rosalia*, juveniles showed a prevalence of 31% (4/13) in females and 26% (5/19) in males, with an overall prevalence of 29% (9/31). Among subadults, females had a prevalence of 17% (2/12), and males had 27% (3/11), with an overall prevalence of 22% (5/23). In adults, females had a prevalence of 43% (10/23), and males had 40% (10/25), with an overall adult prevalence of 42% (20/48). *L. rosalia* individuals were organized into groups at different geographic locations, and, in this study, animals belonging to 12 locations and 31 groups were sampled. The prevalence varied between 0% to 100% in the collection points (Figure 2 and Table 3). Most of the collections happened in the municipality of Silva Jardim—RJ, and one collection happened in Rio Bonito—RJ (Figure 2). Some of the points were well sampled, as were the cases of Afetiva, with 26 animals, and Nova Esperança, with 19 animals, while others were poorly represented, as in the cases of Ribeirão, Sítio Quelinho, and Tertúlio, which had only two animals sampled (Table 3). The mean viral load per location ranged from 2.74 log to 4.56 log (Table 3). The collection points that had the largest number of sampled groups were Afetiva, Nova Esperança, and Rio Vermelho, with six, five, and four groups, respectively (Appendix A). One of Afetiva’s groups, Afetiva 2, was the most sampled, with 12 animals. In total, 31 groups were represented, belonging to 12 locations (Appendix A). The average viral load varied between 1.88 log to 4.34 log (Appendix A).

Information regarding SFV prevalence and viral load according to the location of data collection for *Leontopithecus rosalia* can be found in Appendix A.

### 3.3. SFV Viral Load

We observed a mean viral load of 4.37 log copies per 10^6^ cells (log_10_ 3.03; range: 1.29 log to 5.89 log). In *L. rosalia*, the mean SFV viral load was 4.48 log per 10^6^ cells (log 3.17; range: 1.43 log to max 5.89 log), while in *L. chrysomelas*, the mean SFV viral load was 4.37 log copies per 10^6^ cells (log_10_ 3.03; range: 1.29 log to 3.30 log) (Figure 3A). While the maximum viral load in *L. chrysomela*s was 3.3 log, in *L. rosalia,* 17 individuals had viral loads exceeding 3.3 log, with 12 of them above 3.5 log and 7 above 4. Finally, our linear mixed-effect models revealed that any of the morphological traits (age, sex, or body weight) significantly explained viral load in our studied species (Table 4). However, the random effect of species showed a significant contribution (*p* = 0.007), suggesting differences in the SFV viral load per species.

### 3.4. Phylogenetic and Timescale Analyses

All samples positive for SFV infection in the qPCR diagnosis (*n* = 50), 34 from *L. rosalia* and 16 from *L. chrysomelas*, were subjected to PCR of the larger fragment of *pol*. However, only three samples of *L. chrysomelas* and one sample of *L. rosalia* had that fragment successfully amplified and were thus directed to sequencing. Only one of the positive samples analyzed, a sample from *L. rosalia*, had the sequence successfully determined (MP261), generating a 213 bp DNA fragment (GenBank accession number: PP960560).

The multiple sequences alignment passed tests of phylogenetical signal with satisfactory results. Although 63.1% of quartets in the likelihood mapping analysis were resolved into distinct topologies (Appendix A), a robust phylogenetic signal was evidenced by minimal substitution saturation (Iss < Iss.c, Appendix A) and high alignment confidence (GUIDANCE2 = 0.995, Appendix A). The maximum likelihood phylogeny inferred recovered a topology consistent with previously proposed SFV, with most host-specific viral lineages inferred with high support values (SH-aLRT  > 75, UFBoot > 75). The *L. rosalia*-generated sequence grouped with the SFV of the Cebidae and Callitrichidae families, close to another sequence from *L. rosalia* (SFVlro; GenBank accession number: PP960560) as expected, forming a clade of SFVlro with strong support (SH-aLRT  > 98.8, UFBoot > 100). However, it did not form a sister clade with either of the two circulating *L. chrysomelas* SFV lineages (SFVlchrysom) (Figure 4A). The phylogeny of SFV infecting the Cebidae and Callitrichidae families can be further visualized in Figure 4B.

While our phylogeny inferred the SFV infecting Afro-Eurasian primates as a clade within each of the host families that they infected, we did not observe the same happening in the SFV infecting AP. For instance, the SFV infecting the family Pitheciidae did not form a clade in our analysis. Also, we observed the presence of two lineages of SFV circulating both in the *Sapajus* and *Leontopithecus* genera. Although the SFV families were mixed in the AP, we observed clades in each SFV lineage. The SFV infecting the *Leontopithecus* genus always formed a sister clade of SFV from the Cebidae or Callitrichidae family.

Considering this unique pattern, a timescale phylogenetic tree was obtained using the RelTime-ML method (Figure 5). The SFV found in OWP served as calibration points, using its host divergence dates to calibrate internal nodes of the viral tree. The *L. rosalia* SFV sequence obtained in this work had an origin calculated to be 0.0836 Mya, indicating a recent circulation among *L. rosalia*. The SFVlro shared recent ancestors with *Sapajus* SFV (0.7071 Mya) and *L. chrysomelas* SFV 1 (1.1471 Mya), belonging to an older monophyletic lineage of Cebidae SFV (3.79 Mya). A similar pattern was found for the *L. chrysomelas* SFV 2, which shared an ancestor with *Sapajus nigritus* SFV (2.4733 Mya) and was a sister clade to a lineage of *Callithrix* SFV (3.652 Mya). Interestingly, a pattern of the host switching between Callitrichidae and Cebidae was observed, as the monophyletic lineage of all Callitrichidae SFV and Cebidae SFV (4.2332 Mya) was not exclusive to either host family. Node dates and confidence intervals of major clades of SFV can be found in Appendix A.

## 4. Discussion

Despite the high diversity and broad geographic distribution of American primates [56], data on the prevalence of viral agents, including simian foamy virus (SFV), remain scarce—especially in free-living populations. SFVs are widely disseminated retroviruses known to coevolve with their primate hosts [7]. In this study, we reported the prevalence, the viral load, and the phylogenetic relationship of SFV in free-living populations of *L. chrysomelas* and *L. rosalia*. [8,11,34]. Our results showed that the proviral load was distinct between species, but any of the studied morphological trait (age, body weight, and sex) significantly predicted it. The observed prevalence rates aligned with previous reports, which ranged between 20% and 50% in free-living American primates [8,11,34]. This difference may be explained by the low number of *L. chrysomelas* collected. SFV prevalence in free-living *L. chrysomelas* was comparable to that observed in recently captured captive individuals [11]: the prevalence was similar, either in the comparison of individuals with up to six months of captivity (*p* = 0.72) or those that were in captivity for more than six months (*p* = 0.13). Consistent with findings in both Afro-Eurasian primates and American primates, no significant sex-related differences in SFV prevalence were observed, indicating that sex does not influence transmission [11,34,57].

Hood and collaborators [58] reported that SFV prevalence increases with age in *Macaca fascicularis*, likely due to the chronic nature of SFV infection and cumulative exposure over time [20,59]. To explore this trend, we compared SFV prevalence among juveniles, subadults, and adults. In *L. rosalia*, we found similar results, with the prevalence ranging from 29% in juveniles to 42% in adults. In *L.* chrysomelas, we observed 50% in both juveniles and subadults and 17% in adults. However, the small sample sizes for juveniles (*n* = 2) and subadults (*n* = 4) may have limited the reliability of these estimates. In general, there was a high variability in SFV prevalence among the collection points sampled in Silva Jardim and Rio Bonito. The *L. rosalia* population was distributed in 13 management units, in an area of approximately 4500 km^2^ of lowland Atlantic coastal rainforest. In 2019, AMLD detected 24 social groups [60,61]. The dispersion of *L. rosalia* occurs more frequently from small and nearby fragments than from large, isolated forests. That being so, a fragmented landscape may lead to low dispersal rates [62]. The population dynamics and viability were highly affected by dispersion [61]. The *L. chrysomelas* population of this study was present in a fragmentated habitat, near urban areas in Niterói-RJ, and their interaction with humans and domestic animals was already registered [61].

We attempted to correlate the geospatial arrangement of *L. rosalia* with SFV prevalence, assuming that nearby groups would have a similar prevalence, but due to the small sample size of each group, it was not possible to conduct a robust statistical analysis. With the provided data, we also observed the prevalence in each group within each of the AMLD animal collection points. Some points were better represented than others, as was the case at the Afetiva collection point (Table 4). On the other hand, at other collection points, sampling probably did not represent the local population, such as at the Sítio Quelinho and Andorinha points, with samples from a single group of two and four animals, respectively.

The sensitivity of qPCR was between 10–60% for 1 copy of DNA and 90–100% for 20 copies, comparable with serological tests [34]. For phylogenetic analysis, a conventional PCR of the *pol* region was carried out, and 8% (4 out of 45) of the positive samples for SFV were amplified. Such limited PCR amplification success may be indicative of a great genetic variability of SFV strains circulating among this group of animals. The *pol* PCR was developed with the few available complete sequences of SFV infecting American primates, which included SFV from *Sapajus*, *Callithrix*, and *Leontopithecus* [11]. In this sense, the primers used may not comprise the diversity of SFVs present in these families of primates, a major limitation for this study of molecular characterization by conventional methods. The diversity in the *pol* region could reach 54% when comparing the SFVs infecting different species of American primates [34]. The few available partial *pol* sequences of SFV infecting the *Leontopithecus* genus, only 13, made it difficult to obtain the design of specific primers. Furthermore, the efficiency of the primers used to amplify a larger fragment of *pol* was usually low [8,11]. Besides that, gDNA concentrations in the PCR-positive samples were low (5.9–7.5 ng/μL). In the only successfully sequenced fragment, only the forward primer generated usable data, yielding a 213 bp sequence, and further Sanger sequencing attempts failed. Future studies using high-throughput sequencing may be more effective for these divergent SFV strains. As new strains of SFV were sequenced, an improvement in molecular techniques for detecting this virus was warranted. The only SFV sequence of *L. rosalia* obtained formed a clade with a previous SFVlro sequence from a captive *L. rosalia* specimen from the Rio de Janeiro Primate Center obtained in the study by Muniz and collaborators [20]. However, there was no grouping with any clade of the SFVlcm lineages, which would be expected by the co-divergence hypothesis. The identification of two circulating strains in *L. chrysomelas* was observed in [11], in which viral strains SFVlcm-1 and SFVlcm-2 were identified. In agreement with what was found in Muniz et al. [8], we also observed two clades of SFV infecting the *Sapajus* genus. This recurrent pattern of multiple SFV strains found within a distinct genus of NP can imply a more complex scenario than the expected co-divergence hypothesis. Ghersi et al. [9] considered this as a dynamic co-speciation between SFV and their hosts in AP, with ancient cross-genus transmissions of SFV as the cause of this unique pattern within AP SFV. The sequencing of larger regions of *pol* and/or other viral genes of SFVs infecting AP will allow more robust phylogenetic analyses for the understanding of the evolution of SFV in AP.

Despite the small generated fragment (213 bp), considerations of the dynamics of the evolution of SFV could be investigated using molecular dating techniques. However, since the sequence generated was short compared to the *pol* gene (approximately 3440 bp), testing such hypotheses is challenging, and there must be caution. In our timescale, the SFVlro shared recent ancestors with *Sapajus* SFV (0.7071 Mya) and *L. chrysomelas* SFV 1 (1.1471 Mya), indicating a recent circulation among those species. Even more, all Callitrichidae SFV and Cebidae SFV shared an ancestor strain dated to 4.2332 Mya. Such molecular dating matches with the timescale phylogeny of Ghersi et al. [9], where Callithrix and Cebidae SFVs were dated to ~4.36 Mya (3.08–6.06). Both analyses presented a more recent ancestor than the ones found previously by our group (7.94 Mya 5.17–14.05) [20]. Nevertheless, all time trees seemed like a consensus that the split of this SFV lineage happened much later than the Callithrix and Cebidae host split [~20.70 Ma (95% HPD = 19.02–22.41) [21], possibly representing cross-genus transmission between the hosts. Further studies focused on the diagnosis, genomic sequencing, and evolutionary modeling of SFV naturally infecting wild populations are key to understanding such unique dynamics within the AP SFV.

## 5. Conclusions

In conclusion, we reported the prevalence, the proviral load, and the phylogenetic relationship of SFV in free-living populations of *L. rosalia* and *L. chrysomelas*. We did not observe a significant difference in SFV prevalence, but we did on the proviral load, according to the species. Proviral load was not influenced by sex, age category, or body weight. We also did not observe a relationship between the geographic distribution of samples from *Leontopithecus rosalia* and SFV prevalence. The clustering of SFVlro with the SFV from *Sapajus* may be the result of one or more interspecies transmission events. Further molecular and phylogenetic analyses would be necessary to shed light on this issue.

## Figures and Tables

**Figure 1 viruses-17-01072-f001:**
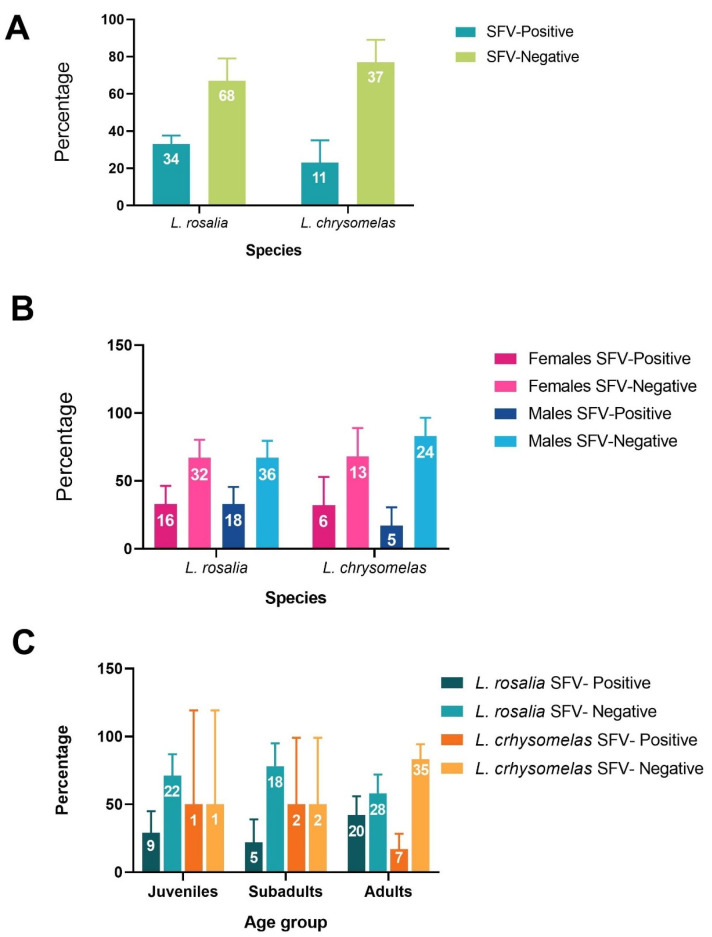
Prevalence of simian foamy virus (SFV) infection according to *Leontopithecus* species (**A**), sex (**B**), and age group (**C**).

**Figure 2 viruses-17-01072-f002:**
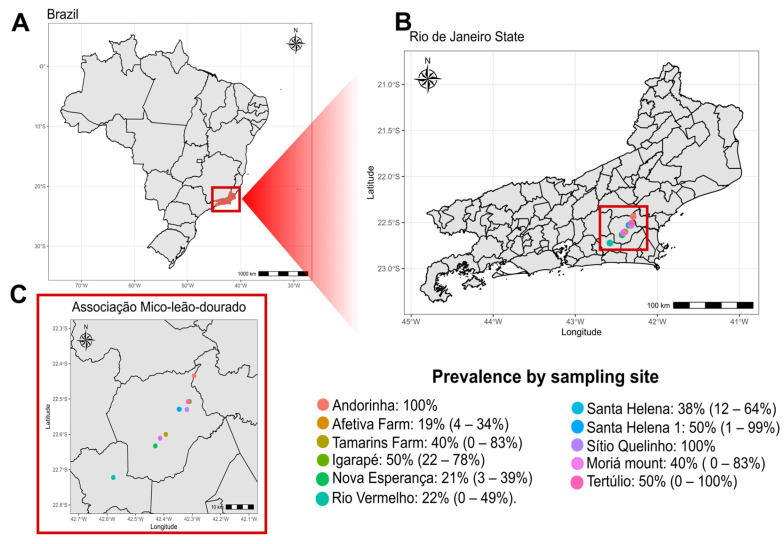
Geographic distribution of groups sampled in the state of Rio de Janeiro (RJ) (**A**), in the municipalities of Silva Jardim and Rio Bonito (**B**). In (**C**), we highlighted the municipalities where the collections were carried out, with Silva Jardim being the most representative (10 collection points) and Rio Bonito with only one point of data collection. Each color represents a different collection point. In the legend, we showed the SFV prevalence in each one of the collection points.

**Figure 3 viruses-17-01072-f003:**
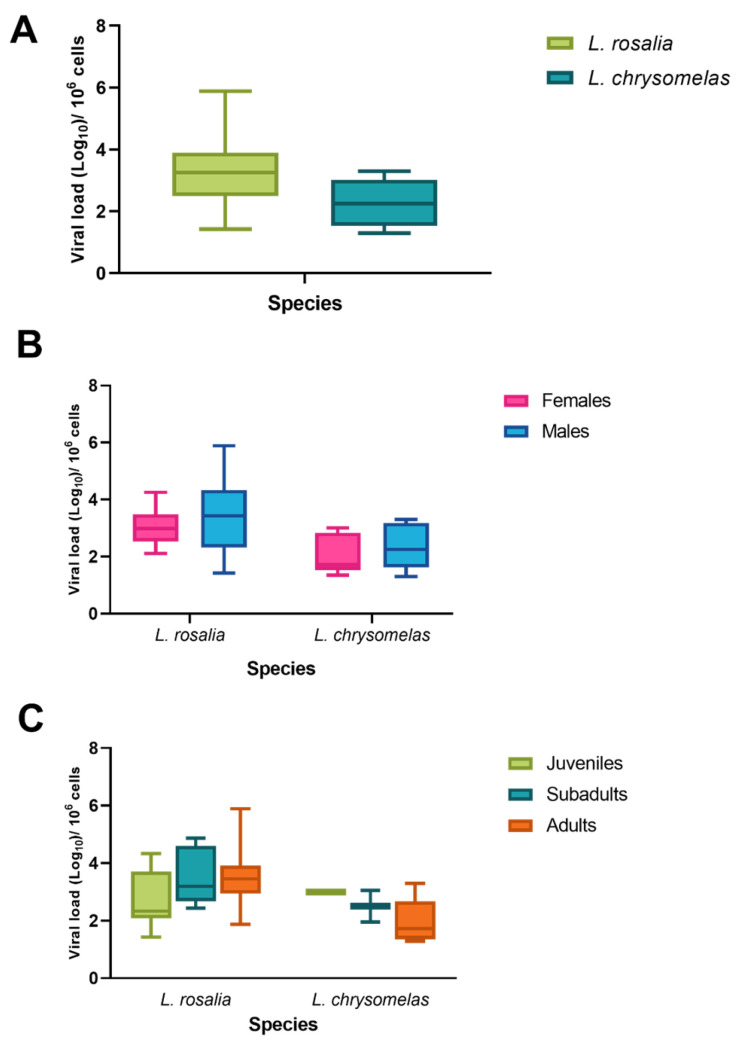
Simian foamy virus (SFV) viral load according to *Leontopithecus* species (**A**), *p*-value = 0.0051; sex (**B**); and age group (**C**).

**Figure 4 viruses-17-01072-f004:**
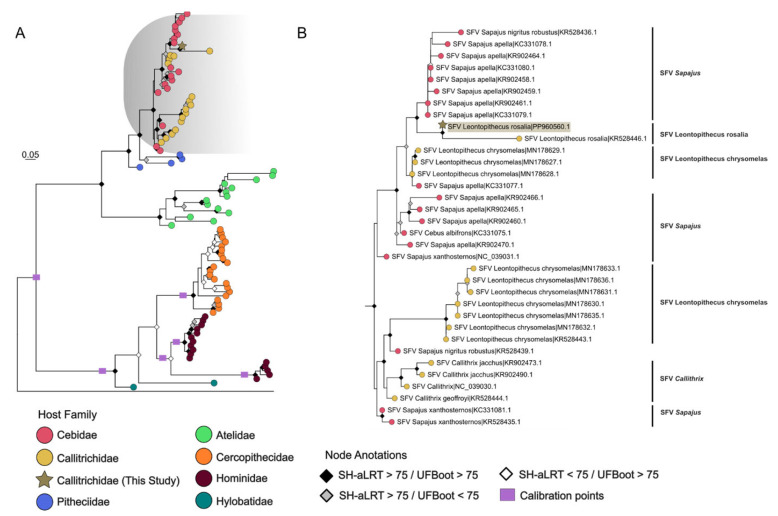
Phylogenetic tree inferred using maximum likelihood analysis with a fragment of SFV viral polymerase (213-bp). The new sequence generated in the current study is marked with a golden star. The host species within the Callitrichidae and the Cebidae families are listed. The node labels are colored according to the host family used in the dataset. Node labels are colored according to the SH-type approximate likelihood ratio test (SH-aLRT) and ultrafast bootstrap (UFBoot) supports. The node labels colored in black represent support for SH-aLRT and UFBoot equal to or greater than 75%. When only SH-aLRT is superior to the cutoff, the label is represented in gray, while only UFBoot node labels are represented by white squares. When both parameters are lower than 75%, no label is depicted. Purple rectangles represent the calibration points used to further explore the timescale of the phylogeny. In (**A**) we see the phylogeny of SFV infecting the five Primate families. (**B**) represents the phylogeny of SFV infecting Cebidae and Callitrichidae families.

**Figure 5 viruses-17-01072-f005:**
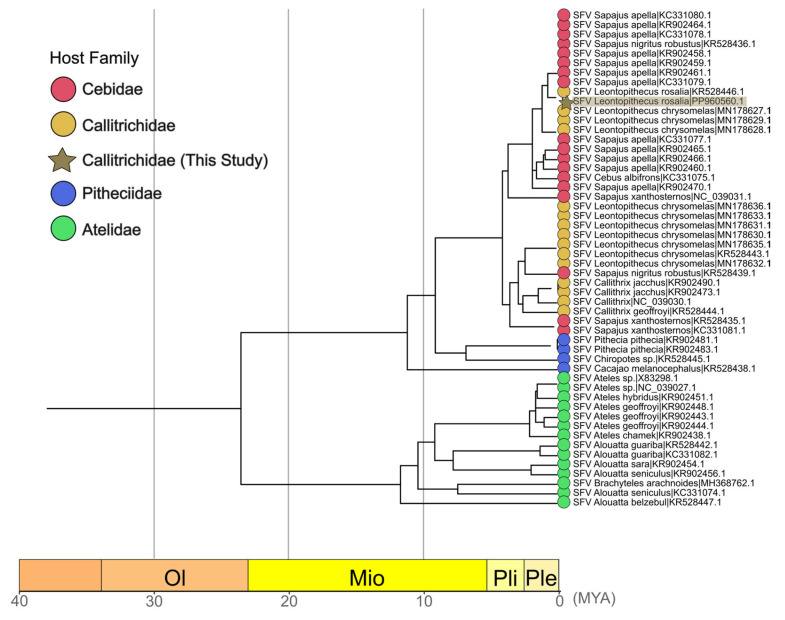
Timescale phylogenetic tree generated by RelTime-ML. Estimated host divergence dates were used to calibrate internal nodes of the viral tree. The node labels are colored according to the host family used in the dataset. The sequence generated in the current study is marked with a golden star. The *x*-axis summarizes the geological timescale of the time tree: Oligocene (Ol), Miocene (Mio), Plioceno (Pli), and Pleistocene (Ple).

**Table 1 viruses-17-01072-t001:** PCR primers for the SFV polymerase (*pol*) sequence.

Round	Primer	Sequence (5′ to 3′)	Direction	Size	Annealing Temperature (35 Cycles)
1st	5960	TACCACTTTGTAGGTCTTCC	Forward	486 bp	53.4 °C
	5878	CTTTGGGGGTGGTAAGG	Reverse		
2nd	5474	GCCAAACATGAGAAAGGATG	Forward	404 bp	54 °C
	5878	CTTTGGGGGTGGTAAGG	Reverse		

**Table 2 viruses-17-01072-t002:** Demographic and morphometric data of *Leontopithecus rosalia* and *Leontopithecus chrysomelas*.

Individuals	*L. rosalia*	*L. chrysomelas*
Total sample size	102	48
Males	54 (53%)	29 (60%)
Females	48 (7%)	19 (40%)
Adults	48 (47%)	42 (88%)
Subadults	23 (23%)	4 (8%)
Juveniles	31 (30%)	2 (4%)
Average weight (grams)	521 (259–754) g	603 (320–740) g
Average number of individuals sampled per site	5 (2–26)	N/A *

N/A *—not available.

**Table 3 viruses-17-01072-t003:** SFV prevalence and viral load according to age category in *Leontopithecus rosalia*.

Collection Point	Animals Sampled	Juveniles	Subadults	Adults	Prevalence (%)	Mean Viral Load *
Afetiva	26	12	9	5	19 (4–34)	3.04
Tamarins	5	0	2	3	40 (0–83)	3.81
Igarapé	12	3	3	6	50 (22–78)	3.30
Nova Esperança	19	4	4	11	21 (3–39)	3.14
Rio Vermelho	9	4	0	5	22 (0–49)	2.74
Ribeirão	2	0	0	2	0	N/A **
Santa Helena	14	2	4	8	36 (11–61)	3.73
Santa Helena I	4	1	1	2	50 (1–99)	4.56
Sítio Quelinho	2	1	0	1	100	3.91
Tertúlio	2	1	0	1	50 (0–100)	3.93
Monte Moriá	5	4	0	1	40 (0–83)	3.63
Andorinha	3	0	0	3	100	4.22

* Mean viral load (log_10_) per 10^6^ cells. ** Not available.

**Table 4 viruses-17-01072-t004:** Parameter estimates from linear mixed-effect models explaining viral load in *Leontopithecus*.

	Estimate	Standard Error	df	t Value	Pr (>|t|)
Intercept	4.41	1.67	29.12	2.65	0.01
Age (Juvenile)	−1.16	0.69	38.11	−1.68	0.10
Age (Subadult)	−0.29	0.53	38.00	−0.55	0.59
Gender (Male)	0.36	0.28	38.00	1.32	0.19
Body Weight	−0.003	0.002	38.03	−1.12	0.27

## Data Availability

The sequence generated in the present work was submitted to Genbank under the accession number PP960560.

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
