# Peer review of "Simian Foamy Virus Prevalence and Evolutionary Relationships in Two Free-Living Lion Tamarin Populations from Rio de Janeiro, Brazil"

_viruses, 2025, doi:10.3390/v17081072_

Round 1
Reviewer 1 Report
Comments and Suggestions for Authors
The authors describe in this submission the molecular prevalence of SFV in wild and captive lion tamarins (Leontopithecus sp.) in Rio de Janeiro using validated assays and oral specimens. They also measure proviral loads (pVL) and perform phylogenetic analyses on one short SFV polymerase (pol) sequence from one invasive lion tamarin in their study with SFV polymerase sequence from other tropical primates. The authors identified SFV prevalences of 23% and 33% in L. chrysomelas and L. rosalia, respectively, but did not observe differences in ages in SFV-infected animals or by sex. The do report finding higher mean pVL in L. rosalia than in L. chrysomelas. The pol sequence obtained from one wild L. rosalia clustered with an SFV from a captive L. rosalia and not with those from L. chrysomelas. Interestingly, the Leontopithecus SFV cluster with SFV from Sapajus species in their analysis. Overall, the manuscript is written well, and the analyses are generally sound. Below are some suggestions to help strengthen the manuscript, including description of some important information missing from the submission.
Major suggestions:
- My main concern is the robustness of the phylogenetic analyses when using such a short sequence (180-213 nucleotides) from a highly conserved SFV gene. To demonstrate there is sufficient signal in the alignment the authors should include results from with the following bioinformatics tools: (a) likelihood mapping plots in IQTree, (b) Guidance 2 (GUIDANCE Server - a web server for assessing alignment confidence score), (c) checking substitution saturation with DAMBE (XiaLab), and (d) TempEst (TempEst). This is especially true when conduction dating analyses.
It would also be helpful to know where the standard deviations for the phylogenetic clock calibrations came from. They don’t seem to be included in the citation provided by the authors. The authors should provide an explanation for the 11.2 Mya calibration date for the Macaca SFV when the reference cited has that node dated to 5.49 Mya. Finally, why did the authors choose to not include calibration dates within the AP families in their analyses. This may help to better define inferred dates within each AP genus.
Please explain why SFV sequences from squirrel monkeys were not included in the analyses.
Please provide a comparison of your phylogenetic relationships and dates with those previously published.
For the dating analysis why do the authors only show the AP tree and not the complete tree with all Old-World monkey and apes SFV?
Minor suggestions (in order of appearance in manuscript):
- Line 20, abstract. Please replace “invasive” with “introduced” to prevent negative associations with endangered Aps.
- Line 27, abstract. Please define acronyms used for SFVlro and SFVlchrysom and use the accepted SFV nomenclature as defined in Khan et al. “Spumaretroviruses - taxonomy and nomenclature update” Virology 2018.
- Line 29, abstract. Please replace “This was the first study...” with “Our study is the first...”.
- Line 54. Please add reference 7 to {9,11,20].
- Line 64. Do the authors mean “familial” instead of “familiar”?
- Line 97. Are adults really < 12 months old?
- Line 125. Please define acronyms first time used, i.e. polymerase for “pol” and insure it is italicized when used throughout the manuscript.
- Line 132. How much DNA is in the 10ul used in the PCR assays and how were the pVL results standardized?
- Lines 143-148. Please provide table with the primer sequences instead of citing them in a separate article.
- Lines 158-161. NC_03902.1 should be NC_039023.1. Also, please provide the common name for Prosimiispumavirus otocrafo. Please check Table S2 as it only contains 73 taxa and not 92 as shown on line 161.
- Line 182. Please add Catarrhini for the node containing the Pan, Pongo and Macaca since that is the calibration date used.
- Lines 186-187. The final alignment and tree files are not in S1.
- Statistical analyses. Please provide cutoff used to define statistical significance.
- Line 205. Please change “body” to physical”.
- Lines 215-216. Please check the totals. Table 1 shows 19 but ratio is 6/18 and 6/18 + 5/29 = 11/47 not 11/48. It is also unclear why there are no age estimates, weights, etc. for the L. chrysomelas if they are captive animals.
- Lines 221-224. Please also investigate the SFV rates in females and males in each age group.
- Lines 228-230. It would be helpful if the authors could describe the data in Table 2 and 3 instead of just showing the data in the Tables. Please insure the % prevalence in the tables match those in Fig. 2. Table 2, please remove percent symbols for all values in the prevalence column and the 0 preceding 3.04 for the pVL for Afetive Farm.
- Fig 2. The red box location in 2B does not match the map in the box in 2C.
- Section 3.3. Please use log10 values and not 104 copies/106 cells for consistency throughout the manuscript.
- Fig. 3c. Green colors are too similar for juveniles and subadults to distinguish these groups in the graph.
- Fig. 4 legend. Please define SH, aLRT, and UFBoot.
- Line 314. May be better to use only MYA for consistency instead of TYA as the scale in Fig. 5 is in MYA. It would also be helpful to provide a supplementary table of the node dates and confidence intervals inferred in this study with estimates from previous studies for comparison and to better follow the text in the results and discussion.
- Figs. 4 and 5. Could remove SFV from taxa names as this is understood. Please explain why SFVssc from Saimiri is included in alignment table (S2) but is not in the phylogenetic trees.
- Line 331. Please remove “s” from “Americas”.
- Lines 338-349, Discussion. The authors provide suggestions to why they observed pVL differences between the two lion tamarin species, including possible sampling biases. However, they fail to explain how their qPCR data was normalized to input DNA concentrations or CytB levels. Another possibility is that the L. chrysomelas that were introduced may be newly infected with SFV from local AP species without subsequent adaption in the new host resulting in lower pVLs as suggested by the authors via IFN-ᵞ inhibition.
- Discussion. Given the L. chrysomelas in this study were captive and their SFVs cluster with Sapajus and Callithrix species, also from captive animals, combined with the very short sequences analyzed, it is difficult to ascertain the evolutionary and co-speciation history of these SFV. Perhaps the three paragraphs provided by the authors in the discussion describing the phylogenetic results can be shortened to reflect the uncertainty in their conclusions.
- Lines 385-394, discussion. The authors should explain why four pol fragments were successfully PCR-amplified but only one was able to be sequenced. And the one sequenced gave only 213-nucleotides in length from a 404-bp amplicon. It would be beneficial to know the sensitivity and specificity of the PCR primers used and if they tried cloning of the PCR products to sequence these PCR amplicons.
- Lines 455-456, conclusions. This sentence can be removed as it is redundant.
29. Supplementary Materials. Table S2. Please replace Cebus with Sapajus as necessary. Table S3 is Table 3 in the manuscript and not the “annotation” dataset which is a separate file in the zip file. Please describe the two different sequence datasets (“annotation” and “alignment”) here, in the methods, and in the figure legends so the reader knows what each was used for. Please check the taxa in the alignment Table S2 with the sequences in the tree in Fig. 4. It appears that some taxa in S2 are not in the tree. In S2 please use the three letter codes for each SFV following Khan et al. as noted above. Please add in both tables S2 and the annotation file if the sequence is from a captive or wild-caught animal.
Author Response
Reviewer 1
The authors describe in this submission the molecular prevalence of SFV in wild and captive lion tamarins (Leontopithecus sp.) in Rio de Janeiro using validated assays and oral specimens. They also measure proviral loads (pVL) and perform phylogenetic analyses on one short SFV polymerase (pol) sequence from one invasive lion tamarin in their study with SFV polymerase sequence from other tropical primates. The authors identified SFV prevalence of 23% and 33% in L. chrysomelas and L. rosalia, respectively, but did not observe differences in ages in SFV-infected animals or by sex. They report finding higher mean pVL in L. rosalia than in L. chrysomelas. The pol sequence obtained from one wild L. rosalia clustered with an SFV from a captive L. rosalia and not with those from L. chrysomelas. Interestingly, the Leontopithecus SFV cluster with SFV from Sapajus species in their analysis. Overall, the manuscript is written well, and the analyses are generally sound. Below are some suggestions to help strengthen the manuscript, including description of some important information missing from the submission.
Response: Thank you very much for your careful review of our manuscript and for your thoughtful comments and suggestions. We sincerely appreciate your positive feedback regarding the quality of the writing and the soundness of our analyses. We carefully addressed each of your suggestions and incorporated the missing information you highlighted to strengthen the manuscript.
Point 1: My main concern is the robustness of the phylogenetic analyses when using such a short sequence (180-213 nucleotides) from a highly conserved SFV gene. To demonstrate there is sufficient signal in the alignment the authors should include results from with the following bioinformatics tools: (a) likelihood mapping plots in IQTree, (b) Guidance 2 (GUIDANCE Server - a web server for assessing alignment confidence score), (c) checking substitution saturation with DAMBE (XiaLab), and (d) TempEst (TempEst). This is especially true when conduction dating analyses.
Response 1: We are grateful for the considerations regarding the phylogenetic signal in our dataset. Such concern is necessary as we are dealing with viruses with high genetic diversity with a small fragment and trying to comprehend its evolutionary history in time. Considering the recommendations, our datasets passed through likelihood mapping plots in IQTree, Guidance 2 and DAMBE with satisfactory levels (Supplementary File S1). The TempEst was not possible to be done as we are dealing with genetic material collected in the present time and the calibration was done considering the host evolutionary history (Kuderna et al., 2024). We acknowledge that 30.8% of quartets in the likelihood mapping analysis remain unresolved, a known challenge for short, conserved sequences. However, the retained phylogenetic signal (Iss < Iss.c, GUIDANCE2 = 0.995) and strong bootstrap support for key nodes (>70%), combined with host-based calibrations, ensure the reliability of our divergence time estimates. We have revised the text to explicitly address this nuance (Page 16, lines 398-403).
Point 2: It would also be helpful to know where the standard deviations for the phylogenetic clock calibrations came from. They don’t seem to be included in the citation provided by the authors.
Response 2: Thank you for your attention to this matter. The standard deviations were calculated to coincide with the ranges available by the reference. To improve clarity, this part was adjusted in the text (page 5, lines 211-227).
Point 3: The authors should provide an explanation for the 11.2 Mya calibration date for the Macaca SFV when the reference cited has that node dated to 5.49 Mya..
Response 3: We appreciate your consideration. The calibration of 11.2 Mya date was an error, the correct calibration indeed is 5.49 Mya as you evidenced. The ML-Timetree was reinfered with the corrected calibration and corresponding information on the article was corrected (page 5, line 217).
Point 4: Finally, why did the authors choose to not include calibration dates within the AP families in their analyses. This may help to better define inferred dates within each AP genus.
Response 4: Thank you for the opportunity to clarify. The points of calibration were chosen in points of our SFV ML phylogeny which mirrors the Primates major splits found by Kuderna (doi:10.1126/science.abn7829). For the Catarrhine clade, major splits among SFV and NHP were similar and were chosen as calibration points. Wherever, for the AP, the points did not match. For instance, the first major split within AP found by Kuderna et al., 2023 is the split of Pitheciidae. However, in our fragment, Atelidae is the first lineage of AP SFV to diverge. Combining with the paraphiletism found within Pitheciidae SFV and the complex scenario reported within the Cebidae and Callitrichidae SFV, we choose to use only the major points within the SFV Catarrhine splits, using the maximum number of calibrations and preferring calibrations closer to the root to an improvement in estimates of divergence times, following recommendations within Duchêne et al., 2014 (doi:10.1016/j.ympev.2014.05.032). If any unnoticed calibration point is available, we are at the disposal to add to our analysis. To improve clarity, this part of the manuscript of calibration node chosen was adjusted in the text (page 5, lines 211-227)
Point 5: Please explain why SFV sequences from squirrel monkeys were not included in the analyses.
Response 5: The SFV sequences from squirrel monkeys (Saimiri spp.) were excluded from the dataset and subsequent analyses due to their well-documented phylogenetic divergence from other simian foamy viruses, as highlighted in prior studies (Santos et al., 2019 doi:10.3390/v11100967, Ghersi et al., 2015 doi: 10.1186/s12977-015-0214-0, Muniz et al., 2015 doi: 10.1186/s12977-015-0217-x). We acknowledge that SFV Saimiri represents an intriguing exception to co-divergence, and its exclusion underscores the need for future work to explore the mechanisms driving its unique evolutionary trajectory.
Point 6: Please provide a comparison of your phylogenetic relationships and dates with those previously published.
Response 6: We now provided a comparison between our phylogenetic relationships and dates with others already published (doi: 10.1186/s12977-015-0214-0) and the results were concordant (page 20, line 550- page 21, line 556).
Point 7: For the dating analysis, why do the authors only show the AP tree and not the complete tree with all Old-World monkey and apes SFV?
Response 7: We are grateful for the consideration. We added only AP in the timetree to emphasize the clade of interest in our findings. A novel figure with AP and OWP is available in Supplementary File S1.
Point 8:
Line 20, abstract. Please replace “invasive” with “introduced” to prevent negative associations with endangered Aps.
Line 27, abstract. Please define acronyms used for SFVlro and SFVlchrysom and use the accepted SFV nomenclature as defined in Khan et al. “Spumaretroviruses - taxonomy and nomenclature update” Virology 2018.
Line 54. Please add reference 7 to {9,11,20].
Line 64. Do the authors mean “familial” instead of “familiar”?
Line 125. Please define acronyms first time used, i.e. polymerase for “pol” and insure it is italicized when used throughout the manuscript.
Lines 143-148. Please provide table with the primer sequences instead of citing them in a separate article.
Lines 158-161. NC_03902.1 should be NC_039023.1. Also, please provide the common name for Prosimiispumavirus otocrafo
Line 182. Please add Catarrhini for the node containing the Pan, Pongo and Macaca since that is the calibration date used.
Statistical analyses. Please provide cutoff used to define statistical significance.
Line 205. Please change “body” to physical”.
Lines 221-224. Please also investigate the SFV rates in females and males in each age group.
Lines 228-230. It would be helpful if the authors could describe the data in Table 2 and 3 instead of just showing the data in the Tables. Please insure the % prevalence in the tables match those in Fig. 2. Table 2, please remove percent symbols for all values in the prevalence column and the 0 preceding 3.04 for the pVL for Afetive Farm.
Fig 2. The red box location in 2B does not match the map in the box in 2C.
Section 3.3. Please use log10 values and not 104 copies/106 cells for consistency throughout the manuscript.
Fig. 4 legend. Please define SH, aLRT, and UFBoot.
Line 331. Please remove “s” from “Americas”.
Lines 455-456, conclusions. This sentence can be removed as it is redundant.
Table S2. Please replace Cebus with Sapajus as necessary.
Response 8: All suggested corrections were carried out. We have also revised the entire manuscript text in search for additional English style improvement.
Point 9: Line 29, abstract. Please replace “This was the first study...” with “Our study is the first...”.
Response 9: As other revisors asked us to avoid statements of primacy, we rewrote the sentence (lines 31-33).
Point 10: Line 97. Are adults really < 12 months old?
Response 10: We sincerely apologize for any confusion caused. We changed the symbol (page 3, lines 119-120).
Point 11: How much DNA is in the 10ul used in the PCR assays and how were the pVL results standardized?
Response 11: The amount of DNA used in the qPCR varied according to the samples, and the range was between 20 and 1.000 ng. To calculate the viral load, we made serial dilutions of a plasmid containing an insert sequence of SFVssc and quantified them, making a standard curve that was used to estimate the concentration of DNA in our samples. To calculate the integrated proviral load in 106 cells, the amount of DNA was converted from nanograms to picograms (pg) and multiplied by 1 million cells, with one cell having on average 6 pg of DNA (TANDON R, et al., 2005, doi:10.5167/uzh-3037).
Point 12: Please check Table S2 as it only contains 73 taxa and not 92 as shown on line 161.
Response 12: Thank you very much for the observation. Our first dataset was made up of 73 taxa, but we improved our dataset to 92 taxa and we didn’t have corrected all tables. Now, the data are matching. Information regarding our dataset can be found in Supplementary File 1, Supplementary Table 1.
Point 13:Lines 186-187. The final alignment and tree files are not in S1.
Response 13: The final final alignment and tree files are available in the github repository, in the following link:
https://github.com/deagirardi/Supplementary_data_SFV_L.rosalia_L.chrysomelas
Point 14: Lines 215-216. Please check the totals. Table 1 shows 19 but ratio is 6/18 and 6/18 + 5/29 = 11/47 not 11/48. It is also unclear why there are no age estimates, weights, etc. for the L. chrysomelas if they are captive animals.
Response 14: We apologize for the oversight. We have corrected the number of females mentioned in the text. Additionally, there was an error in the table regarding age estimates and weights, which has now been addressed. The L. chrysomelas individuals were free-living primates captured in the wild, and the samples were collected at the moment of capture. However, our collaborators did not record the GPS coordinates, and the height measurements were taken using a different method than for L. rosalia. Due to these inconsistencies, we were unable to make direct comparisons between the species. Therefore, we decided to remove the height data from the table, as it was not used in any of our analyses. The corrected age groups and weight data for L. chrysomelas are now presented in Table 2 (page 6-7, lines 262-263).
Point 15: Fig. 3c. Green colors are too similar for juveniles and subadults to distinguish these groups in the graph.
Response 15: Thanks for observation. We changed the colors to green, blue and orange instead of shades of green, to make the graph easier to understand.
Point 16: Line 314. May be better to use only MYA for consistency instead of TYA as the scale in
Fig. 5 is in MYA. It would also be helpful to provide a supplementary table of the node dates and confidence intervals inferred in this study with estimates from previous studies for comparison and to better follow the text in the results and discussion.
Response 16: Thanks for the consideration. Such changes were made (page 18, lines 422-423) and the table was made available in supplementary file (Supplementary File S1, Table S4) to greater clarity.
Point 17: Figs. 4 and 5. Could remove SFV from taxa names as this is understood. Please explain why SFVssc from Saimiri is included in alignment table (S2) but is not in the phylogenetic trees.
Response 17: Thank you for the considerations. To avoid miscommunication regarding our phylogeny corresponding to the SFV, not its hosts, we prefer to maintain the SFV in taxa name. Regarding SFVssc, this sequence is present in the alignment table by mistake as this sequence was previously removed from the dataset, as explained previously.
Point 18: Lines 338-349, Discussion. The authors provide suggestions to why they observed pVL differences between the two lion tamarin species, including possible sampling biases. However, they fail to explain how their qPCR data was normalized to input DNA concentrations or CytB levels. Another possibility is that the L. chrysomelas that were introduced may be newly infected with SFV from local AP species without subsequent adaption in the new host resulting in lower pVLs as suggested by the authors via IFN-ᵞ inhibition.
Response 18: We tried to explain differences in viral loads at an individual level and failed to explain it at a species level. The mode of sampling collection was the same for both species, but some articles discuss that the distributions of SFV infected cells are not homogeneous in the oral captivity, with them being localized in sparse loci (Falcone, et al., 1999, doi: 10.1006/viro.1999.9634). We argue that this phenomenon could explain differences in viral loads, where the infected cells could or not be taken in the swab sampling. Nevertheless, that could explain these differences only on an individual level, not at species level. With the multivariate analysis, we discovered that besides the significance seen in the proviral load in different species, any of our variables (species, age, weight or sex) explain our difference (page 20, lines 464-468).
Point 19: Discussion. Given the L. chrysomelas in this study were captive and their SFVs cluster with Sapajus and Callithrix species, also from captive animals, combined with the very short sequences analyzed, it is difficult to ascertain the evolutionary and co-speciation history of these SFV. Perhaps the three paragraphs provided by the authors in the discussion describing the phylogenetic results can be shortened to reflect the uncertainty in their conclusions.
Response 19: Thank you for your consideration. In truth, the L. chrysomelas are not captive animals. We sincerely apologize for any confusion caused. They are free-living animals that were captured (the sampling occurred in this first contact with them) and, after that they are maintained in an enclosure until its transportation for their natural habitat in another state (Bahia). On lines 79-88 from introduction we explained better the mode of living of L. chrysomelas. On lines 112–115 from methodology, we also elucidate in which conditions the collections were made. We agree that the sequence is too short and, because of that, we must be careful about the assumptions we make. We rewrote the paragraphs to make them concise and clearer.
Point 20: Lines 385-394, discussion. The authors should explain why four pol fragments were successfully PCR-amplified but only one was able to be sequenced. And the one sequenced gave only 213-nucleotides in length from a 404-bp amplicon. It would be beneficial to know the sensitivity and specificity of the PCR primers used and if they tried cloning of the PCR products to sequence these PCR amplicons.
Response 20: Only a small fraction of our samples was amplified by PCR and only one of them was successfully sequenced. That can be a consequence of the design of the primers, that were constructed taking in account all complete sequences of SFV infecting American Primates (five sequences), as we don’t have the complete genome of SFV infecting the Leontopithecus genus and only two partial sequences of pol of SFV infecting L. rosalia and 20 partial pol sequence of SFV infecting L. chrysomelas. The qPCR has a sensitivity that ranges from 10 - 60% for 1 copy of DNA to 90 - 100% for 20 copies, being comparable with immunological tests (Muniz et al., 2017, doi: doi:10.1371/journal.pone.0184251). The primers used to amplify a larger fragment of pol usually have a low efficiency to amplify SFV from Leontopithecus (Miranda et al., 2019, doi: doi:10.3390/v11100931.; Muniz et al., 2013, doi: 10.1371/journal.pone.0067568.; Muniz et al., 2017, doi: doi:10.1371/journal.pone.0184251). And the diversity in pol region can reach 54% when comparing SFV infecting different species of American Primates (Muniz et al., 2013, doi: 10.1371/journal.pone.0067568.).
Point 21: Table S3 is Table 3 in the manuscript and not the “annotation” dataset which is a separate file in the zip file. Please describe the two different sequence datasets (“annotation” and “alignment”) here, in the methods, and in the figure legends so the reader knows what each was used for. Please check the taxa in the alignment Table S2 with the sequences in the tree in Fig. 4. It appears that some taxa in S2 are not in the tree. In S2 please use the three letter codes for each SFV following Khan et al. as noted above. Please add in both tables S2 and the annotation file if the sequence is from a captive or wild-caught animal.
Response 21: Thank you for your comment. We actually had just one dataset that was improved, and ran off our attention that the files were duplicated (one with the previous dataset and another with the new one). We corrected the archive and added if the sequences were related to captive or free-living animals.
Reviewer 2 Report
Comments and Suggestions for Authors
Girardi et al. present a study of SFV prevalence and viral load in two species of NWM (Lion tamarins) living in the same state of Brazil, but at different localization. They report significant prevalence (around ¼ to 1/3 of sampled animals). Prevalence was not related to sex or age. Viral loads in the buccal cavity were overall in the same range in both species. PCR amplification of a longer fragment of the pol gene was performed with the objective of obtaining novel sequences and performed phylogenetic studies. Unfortunately, a single sequence was obtained from the testing of 50 samples. This sequence was derived from L. Roselia and formed a monophyletic clade with another SFVlro sequence. Phylogenetic studies support that the evolutionary pattern of SFV infecting NWM is different from the one of SFV infecting OWM and Apes.
The project is well conceived and realized and data support the conclusions. The manuscript is written in an appropriate way but would benefit from careful proof-reading and use of a more concise style. Some points need to be clarified for readers who are not primatologists while results on viral load deserve a more thorough discussion.
Comments (in order of appearance)
Consistent use of abbreviations across publications: American primates (APs) were called neotropical primates (NPs) in previous publications from the group. New world monkey (NWM) is most commonly used in the field and this term may be adopted.
Line 20 : L. chrysomelas species is qualified as invasive and endangered. The two adjectives are contradictory.
Line 20 : An important information is missing : are the animal captive, captive but free-ranging or wild?
Line 22 : place of collection (Niteroi and Silva gardim) is not necessary in an abstract.
Line 25 : replace average by mean
Line 25 (and line 338): viral load differences are modest, i.e. less than two fold (1071 vs 1862 copies/mL). Is this difference physiologically relevant? In a previous paper (Miranda et al., Viruses, 2019, the VL was 1.7 log higher in L. chrysomela. How do the authors explain the difference between both studies?
Line 26 : The sentence is very generic and misleading. Specify that one novel sequence from a pol gene fragment was obtained.
Line 54 : inadequate usage of the word “theory”.
Line 91-92 : the authors state that “the animals from Niteroi are allocated in enclosures”; On lines 29 and 81, animals are qualified as “free living”. Both statements are contradictory.
Line 87, the authors state that animals are habituated to human contact. Line 94, they state that animals have identification number. Overall, the captivity vs wild status of the animals is confusing for those who are not experts in primatology. The authors should explain in the introduction the living mode of the primates they study in Brazil, and how interactions between animals leading to SFV transmission are different in captivity and for wild animals. Then they should describe the status of the two colonies they have studied.
Line 123 : the qPCR assay cannot distinguished between integrated and unintegrated SFV DNA. The word integrated should be suppressed.
Lines 145-147 : the statement on primers specific for the Cebidae and Callitrichidae is ambiguous as the study is carried out on samples from Callitrichidae only. What was the purpose of amplifying pol genes from SFV infecting another family of NWM? The sequence of the primers may be indicated in the text.
Line 169 : suppress “American”;
Line 201, 203 : replace “n:” by “n=”
Line 221, and line 226 : “it was possible” may be removed. In the whole manuscript, the authors may seek to write shorter sentences to enhance the readability.
Figure 2 and Table 3. The authors should present the SFV prevalence data from both Leontopithecus species in a similar way. The figure is more readable than the table.
Line 258. Replace “proviral” by “viral” because the qPCR assay used cannot distinguished between integrated and unintegrated SFV DNA.
Line 259-267 : one important difference between viral load from both species is that maximal values that are higher in L. Rosalia than in L. Chrysomelas. The authors should described this particularity : number of animals with higher viral load (defined as >3.5 or 4 log) and their sex ad age distribution.
Line 271 : colors on panel B are pink and blue, not green.
Line 344 : The sentence mentioning IFN-g only is misleading as type I IFN, IFN-induced restriction factors and antibodies were also shown to play a role in the control of SFV replication. In addition, in the buccal cavity, type III IFN and antibodies may play a dominant role, while IFN-g and type I IFN may play a modest role.
Line 374-375 : can the hypothesis be verified for the present study?
Line 376-384. After this paragraph, a description of geospatial characteristics of L. chrysomelas colonies should be provided.
Line 398 : use the SFV nomenclature from Kahn et al.
Line 400-402 : the meaning of the sentence is unclear. The authors do not provide information on SFV circulating in sapajus in this paper; they only use their previous data to built a phylogenetic tree with one novel strain.
Comments on the Quality of English Language
see above
Author Response
Reviewer 2
Girardi et al. present a study of SFV prevalence and viral load in two species of NWM (Lion tamarins) living in the same state of Brazil, but at different localization. They report significant prevalence (around ¼ to 1/3 of sampled animals). Prevalence was not related to sex or age. Viral loads in the buccal cavity were overall in the same range in both species. PCR amplification of a longer fragment of the pol gene was performed with the objective of obtaining novel sequences and performed phylogenetic studies. Unfortunately, a single sequence was obtained from the testing of 50 samples. This sequence was derived from L. rosalia and formed a monophyletic clade with another SFVlro sequence. Phylogenetic studies support that the evolutionary pattern of SFV infecting NWM is different from the one of SFV infecting OWM and Apes.
The project is well conceived and realized and data support the conclusions. The manuscript is written in an appropriate way but would benefit from careful proof-reading and use of a more concise style. Some points need to be clarified for readers who are not primatologists while results on viral load deserve a more thorough discussion.
Response: Thank you for your positive assessment of our study and for your thoughtful suggestions and comments. We are pleased that you find the project well conceived, the data supportive of our conclusions, and the manuscript generally appropriate in tone. We carefully addressed each of your suggestions and incorporated the missing information you highlighted, as detailed below.
Point 1: Consistent use of abbreviations across publications: American primates (APs) were called neotropical primates (NPs) in previous publications from the group. New world monkey (NWM) is most commonly used in the field and this term may be adopted.
Response 1: We agree that the nomenclatures should be consistent, and included in our text that the American Primates were formally known as Neotropical Primates. However, some primatologists, anthropologists and ecologists have raised the issue that these terms may have a colonialist bias and that we should start to use new terms. Some recent articles, including Benzanson et al. (2024, doi: 10.1007/s10329-023-01104-6), Adame (2023, doi: 10.1038/d41586-023-00992-4), recommend the use of American or Afroeurasian Primates instead of Neotropical/ New World and Old World Primates, respectively.
Point 2: Line 20 : L. chrysomelas species is qualified as invasive and endangered. The two adjectives are contradictory.
Response 2: Thanks for the observation. We adjusted the sentence and, in page 2, lines 79 -88 of introduction, we described better the conditions of the species.
Point 3: Line 20 : An important information is missing : are the animal captive, captive but free-ranging or wild?
Line 22 : place of collection (Niteroi and Silva Jardim) is not necessary in an abstract.
Line 25 : replace average by mean
Line 26 : The sentence is very generic and misleading. Specify that one novel sequence from a pol gene fragment was obtained.
Line 54 : inadequate usage of the word “theory”.
Line 123 : the qPCR assay cannot distinguished between integrated and unintegrated SFV DNA. The word integrated should be suppressed.
Line 169 : suppress “American”;
Line 201, 203 : replace “n:” by “n=”
Line 221, and line 226 : “it was possible” may be removed. In the whole manuscript, the authors may seek to write shorter sentences to enhance the readability.
Line 258. Replace “proviral” by “viral” because the qPCR assay used cannot distinguished between integrated and unintegrated SFV DNA.
Line 259-267 : one important difference between viral load from both species is that maximal values that are higher in L. rosalia than in L. chrysomelas. The authors should described this particularity : number of animals with higher viral load (defined as >3.5 or 4 log) and their sex ad age distribution.
Line 271 : colors on panel B are pink and blue, not green.
Line 398 : use the SFV nomenclature from Kahn et al.
Response 3: All suggested changes have been made..
Point 4: Line 25 (and line 338): viral load differences are modest, i.e. less than two fold (1071 vs 1862 copies/mL). Is this difference physiologically relevant? In a previous paper (Miranda et al., Viruses, 2019, the VL was 1.7 log higher in L. chrysomelas. How do the authors explain the difference between both studies?
Response 4: Although the difference in the average proviral load is not high, maximal values are higher in L. rosalia than in L. chrysomelas, which could explain the significant difference observed. The animals evaluated in Miranda, et al. (2019) were recently-captured (that means being in an enclosure for up to six months), while in our work we are describing SFV status of infection in free-living animals, with sampling occurring at the moment of capture.
Point 5: Line 91-92 : the authors state that “the animals from Niteroi are allocated in enclosures”; On lines 29 and 81, animals are qualified as “free living”. Both statements are contradictory.
Response 5: We regret any confusion this may have caused. The animals captured in Niterói-RJ, L. chrysomelas were free-living. This species is considered as invasive in Niterói. The Centro de Primatologia do Rio de Janeiro (CPRJ) is undertaking a project involving the capture of these animals, their monitoring and management in captivity, and their subsequent transport and release to their natural habitat, in Bahia State. The sampling was made at the moment of the capture of these animals. Like this, on page 1, lines 21-22 we specified that L. chrysomelas is an invasive free-living primate and on page 2, lines 79-88 we tried to explain better the condition of these animals.
Point 6: Line 87, the authors state that animals are habituated to human contact. Line 94, they state that animals have identification number. Overall, the captivity vs wild status of the animals is confusing for those who are not experts in primatology. The authors should explain in the introduction the living mode of the primates they study in Brazil, and how interactions between animals leading to SFV transmission are different in captivity and for wild animals. Then they should describe the status of the two colonies they have studied.
Response 6: Again we sincerely apologize for any confusion caused. The L. rosalia are free-living primates that are constantly monitored by AMLD, since its foundation. The AMLD is a worldwide example of success of a conservation program. We added some information regarding the ecology of both Leontopithecus species in order to clarify their status in the introduction (page 2, lines 68 - 88).
Point 7: Lines 145-147 : the statement on primers specific for the Cebidae and Callitrichidae is ambiguous as the study is carried out on samples from Callitrichidae only. What was the purpose of amplifying pol genes from SFV infecting another family of NWM? The sequence of the primers may be indicated in the text.
Response 7: The primers were designed for the SFV infecting family Cebidae, before its split between Cebidae and Callitrichidae, using all five complete SFV genomes infecting American Primates available. We didn’t design a primer specific for SFV infecting the genus Leontopithecus because there isn’t either the complete genome of SFV or a greater number of partial sequences of pol of SFV available. This problem could be solved by massively parallel sequencing technologies. We now indicated the sequence of the primers in table 1 (pages 4-5, lines 178-180).
Point 8: Figure 2 and Table 3. The authors should present the SFV prevalence data from both Leontopithecus species in a similar way. The figure is more readable than the table.
Response 8: We feel sorry for that, but unfortunately the collaborators from CPRJ, which provided us the samples from L. chrysomelas, didn't register the geospatial localization in which the animals were captured, so we are not able to create a similar image for both species.
Point 9: Line 344 : The sentence mentioning IFN-g only is misleading as type I IFN, IFN-induced restriction factors and antibodies were also shown to play a role in the control of SFV replication. In addition, in the buccal cavity, type III IFN and antibodies may play a dominant role, while IFN-g and type I IFN may play a modest role.
Response 9: As we wanted just to illustrate that the immunological system may play a role in the proviral load, we decided to remove this sentence, which could be misleading.
Point 10: Line 374-375 : can the hypothesis be verified for the present study?
Response 10: Unfortunately, with the presented data, we can not affirm if or how the presence of roads or traffic led to losses in population size in each one of the collection points. Consequently, we decided to remove this paragraph of the text.
Point 11: Line 376-384. After this paragraph, a description of geospatial characteristics of L. chrysomelas colonies should be provided.
Response 11: As we don’t have the exact geospatial localization, we can only affirm that they live in a fragmented habitat that is close to urban areas. We added that information in the paragraph (lines 508-510).
Point 12: Line 400-402 : the meaning of the sentence is unclear. The authors do not provide information on SFV circulating in sapajus in this paper; they only use their previous data to built a phylogenetic tree with one novel strain.
Response 12: We agree that the sentence was not clear. We identified two clades of SFV infecting the Sapajus genus, but we are not able to confirm that they are two strains. We changed the sentence and specified that there are two clades, instead of two circulating strains (Page 21, lines 542-543).
Reviewer 3 Report
Comments and Suggestions for Authors
Overall comments
This manuscript describes a study of the prevalence and evolution of simian foamy viruses in two species of South American primates. The authors are correct in their assessment that studies of SFV ecology in South America have lagged behind those in Africa, so this study is a welcome addition to the literature. The manuscript is well organized and generally well written, although it would benefit from another round of English language editing.
One overarching concern is that the data on which certain analyses and conclusions are based are scarce. For example, phylogenetic analyses are based on a 213 bp fragment of the integrase gene of one SFV that was successfully amplified and sequenced out of 50 positive samples, and the sequence does not cover the full length of the amplicon. This is a low rate of sequencing success, and it reduces confidence in the quality of the data. It makes the reader wonder if there were a lot of false positive qPCRs and if the sequence data were messy and therefore prone to errors. Similarly, it’s unclear why most morphometric and demographic data are missing for L. chrysomelas (Table 1). Again, this makes the reader suspicious about the quality of the data.
There are also concerns about some of the statistical analyses. In the section on SFV prevalence, no 95% confidence intervals are given in the text or tables 2 and 3, there are no error bars on the graphs in Figure 1, and there are no 95% confidence intervals in the “Prevalence by sampling site” section of Figure 2. There are many good ways to calculate 95% confidence intervals and standard errors for prevalence data. The authors should choose the best one for their data and present these estimates every time they include a prevalence estimate or a graph representing prevalence.
Also, the manuscript contains many “floating p values” in the text, figure legends, and other places. These are p values presented without test statistics or degrees of freedom. Please report full statistical results in all cases. For example, “Student’s t = 5.883, 18 degrees of freedom, 2-tailed p = 0.003.”
Finally, why were multivariate statistical analyses not used to investigate the relationship between species, sex, age group and other factors and viral infection status or load? Lines 188-195 describe simple univariate tests, but these would not be the best choice because predictor variables may be intercorrelated. Approaches such as multivariate linear regression and general linear models might be better for assessing the individual and combined effects of predictors on outcomes.
Overall, this is a well-conceived and well-presented study that would be of value to the field, but issues with data quality and analyses create uncertainty about its internal validity.
Specific comments
Lines 26-27. “Monophyletic” and “clade” are redundant. Change to “within a SFVlro clade.”
Lines 28-30 and throughout the manuscript. Avoid statements of primacy (“first”) and self-congratulation (“great importance”). This sentence is also general and not informative; try writing a final sentence that’s more focused on the precise implications of the results.
Line 34. What does “complex” mean here? Either specify or delete.
Line 67. I believe “natural of” should be “native to.”
Line 69. Change to “mid-1990s.”
Line 69-70. Delete sentence beginning “Therefore.” It’s redundant.
Line 71. Change “for” to “to.”
Line 80. Again, avoid statements of primacy (“first time”). Maybe there is another paper in press right now. Or, maybe somebody did a study but it is published in an obscure venue. Suggestion: change “for the first time” to “new data on.”
Line 92. What does “allocated” mean here? I think another word might be clearer.
Line 108. Change “eppendorf tubes” to “microcentrifuge tubes” and give the manufacturer and location. “Eppendorf” is the name of a specific company.
Line 145. Please provide all primer sequences.
Line 158. Change “to further evolutionary contextualization” to “for subsequent evolutionary analyses.”
Line 169. Change “golden lions tree” to something clearer.
Lines 174-184. How were these calibration dates chosen?
Line 206. 83.1 millimeters is a length, not a weight. Length measurements were not described in the methods, so it is unclear what this length refers to. Knee-heel distance as in Table 1, maybe? Please add details to methods and clarify.
Line 205. How was body condition assessed? If this statement is based on a formula including weight and length, please describe in the methods and add body condition score data to Table 1.
Table 1. Change title to “Demographic and morphometric data…”
Table 1. What does “Median collection per site” mean? Also, why are so many values for L. chrysomelas not available? This should be explained, because it suggests a problem with the field data (see general comments, above).
Lines 259-260. Does the first sentence of section 3.3 mean that all samples yielded data (i.e. that no samples failed)? It’s not clear what is meant by “was possible to measure.”
Lines 276-278. Delete this sentence because it is already in the Methods.
Lines 282-284. This is a very low rate of sequencing success. The authors should carefully explain why sequencing failed for 49/50 samples. The authors should also explain why only 213 bp of sequence was obtained from a 404 bp fragment of integrase. As written, this section makes the reader suspicious that the samples that failed to sequence were false positives, perhaps due to nonspecific amplification during the qPCR. Also, failure to sequence the full amplicon (414 bp should be easy to sequence) makes the reader wonder about sequence quality and sequencing error. The authors should prevent convincing evidence that these issues were not the case.
Lines 333-335. There are problems with the phrase “and their diagnosis and monitoring can serve as a biomarker of zoonotic transmission of the virus between different callitrichid species.” First, “zoonotic” means from animals to humans. Also, how are diagnosis and monitoring a biomarker? There are also no references supporting this statement. I suggest deleting it or re-writing it to be clearer and supported by references.
Lines 335-336. Again, avoid statements of primacy.
Lines 340-349. These are very good thoughts. But, without a multivariate statistical analysis, it’s difficult to know if the differences between the species are “real” or if they are explained by morphometric and demographic differences between the sampled animals of the two species.
Lines 360-362. 42% is much higher than 29% or 22%, so why do the authors conclude that there’s no difference? The prevalence seems higher in adults.
Lines 363-364. Including 95% confidence intervals in the results would help determine whether prevalence values fall into categories (non-overlapping confidence intervals).
Line 386. Typo: SUch.
Line 393. I believe the authors mean “massively parallel sequencing technologies.”
Line 395 and 426 and 430 and 432 and 433 (and maybe elsewhere). “Monophyletic clade” is redundant. Just say “clade.”
Lines 399-408. These are interesting ideas!
Line 411. Change “discarded” to “discounted.”
Line 444. Again, avoid statements of primacy.
Line 452. “zoonotic” means from animals to humans.
Lines 454-456. The two concluding sentences are not clear, and they are results rather than conclusions.
Comments on the Quality of English LanguageCertain sections have awkward or confusing grammar and wording, and I think this may be due to English Language concerns. Other sections do not have this problem, however. The authors should go through the entire manuscript to make sure all sentences are grammatically correct and that vocabulary is appropriate.
Author Response
Reviewer 3
This manuscript describes a study of the prevalence and evolution of simian foamy viruses in two species of South American primates. The authors are correct in their assessment that studies of SFV ecology in South America have lagged behind those in Africa, so this study is a welcome addition to the literature. The manuscript is well organized and generally well written, although it would benefit from another round of English language editing.
Response: We would like to thank the reviewer for their positive and encouraging comments regarding our study. We appreciate the recognition of the relevance of investigating SFV ecology in South American primates and are pleased that our work is considered a valuable addition to the literature. We carefully addressed each of your suggestions and incorporated the missing information you highlighted to strengthen the manuscript. Your input was extremely valuable and will certainly improve the clarity and quality of our work.
Point 1: One overarching concern is that the data on which certain analyses and conclusions are based are scarce. For example, phylogenetic analyses are based on a 213 bp fragment of the integrase gene of one SFV that was successfully amplified and sequenced out of 50 positive samples, and the sequence does not cover the full length of the amplicon. This is a low rate of sequencing success, and it reduces confidence in the quality of the data. It makes the reader wonder if there were a lot of false positive qPCRs and if the sequence data were messy and therefore prone to errors.
Response 1: We understand the concerns about our results. Our PCR was developed taking in account all genomes of SFV infecting American primates available and may not encompass the diversity of all the other American primates species. We don’t have the complete genome of SFV infecting the Leontopithecus genus and only two partial sequences of pol of SFV infecting L. rosalia and 20 partial pol sequences of SFV infecting L. chrysomelas. In fact, in one of our works it was observed that the amplification of a larger fragment of pol usually has a low efficiency to amplify SFV from Leontopithecus (Miranda et al., 2019, doi:10.3390/v11100931.; Muniz et al., 2013, doi: 10.1371/journal.pone.0067568; Muniz et al., 2017, doi:10.1371/journal.pone.0184251). Despite that, the sensitivity of the qPCR is comparable with serological tests (Muniz et al., 2017, doi:10.1371/journal.pone.0184251), being able to detect 10 to 60% of infected American primates for 1 copy of DNA and 90 to 100% for 20 copies (Muniz et al., 2017, doi:10.1371/journal.pone.0184251).
Point 2: Similarly, it’s unclear why most morphometric and demographic data are missing for L. chrysomelas (Table 1). Again, this makes the reader suspicious about the quality of the data.
Response 2: Thank you for bringing this to our attention. We identified an error in the table and added the data regarding the age group of L. chrysomelas, as well as their body weight. Sample collection was conducted in collaboration with distinct research teams, each responsible for a different species. However, our collaborators did not record the GPS coordinates, and the height measurements were taken using a different method than for L. rosalia. Due to these inconsistencies, we were unable to make direct comparisons between the species. Therefore, we decided to remove the height data from the table, as it was not used in any of our analyses. The corrected age groups and weight data for L. chrysomelas are now presented in Table 2 (page 6-7, lines 262-263).
Point 3: There are also concerns about some of the statistical analyses. In the section on SFV prevalence, no 95% confidence intervals are given in the text or tables 2 and 3, there are no error bars on the graphs in Figure 1, and there are no 95% confidence intervals in the “Prevalence by sampling site” section of Figure 2. There are many good ways to calculate 95% confidence intervals and standard errors for prevalence data. The authors should choose the best one for their data and present these estimates every time they include a prevalence estimate or a graph representing prevalence. Also, the manuscript contains many “floating p values” in the text, figure legends, and other places. These are p values presented without test statistics or degrees of freedom. Please report full statistical results in all cases. For example, “Student’s t = 5.883, 18 degrees of freedom, 2-tailed p = 0.003.”
Response 3: We calculated the confidence intervals and included them in our texts and tables. We also added error bars in our graphs. We calculated the confidence interval in the “Prevalence by sampling site” section of Figure 2, with the exception of those sites who had a prevalence of 100% and, therefore, it was not possible to calculate it. We also added the missing information regarding the p values.
Point 4: Finally, why were multivariate statistical analyses not used to investigate the relationship between species, sex, age group and other factors and viral infection status or load? Lines 188-195 describe simple univariate tests, but these would not be the best choice because predictor variables may be intercorrelated. Approaches such as multivariate linear regression and general linear models might be better for assessing the individual and combined effects of predictors on outcomes. Overall, this is a well-conceived and well-presented study that would be of value to the field, but issues with data quality and analyses create uncertainty about its internal validity.
Response 4: Thank you, for your suggestion. We performed multivariate analyses as requested and observed that the differences in proviral load are significant, but the significance observed is not explained by any of our variables. We explain how the multivariate analysis was made in the methodology (page 6, lines 233-239). On table 3 (page 10, lines 331-332), we describe the p-values, the 95% confidence interval and the odds ratio of the prevalence according with species, sex, age group and weight, assed by Binary Logistic Regression. Through this analysis we also found that the difference in the prevalence is significant according to the species and to the weight (Page 20, lines 465-466). On figure 4 (page 16), we made available a graph of the result of the Multiple Linear Regression comparing the viral load/per 106 cells by Species, Age Group, Weight and Sex. We found a significant difference when we compare the viral load in both species, but it was not explained by any of the variables used.
Point 5:
Lines 26-27. “Monophyletic” and “clade” are redundant. Change to “within a SFVlro clade.”
Lines 28-30 and throughout the manuscript. Avoid statements of primacy (“first”) and self-congratulation (“great importance”). This sentence is also general and not informative; try writing a final sentence that’s more focused on the precise implications of the results.
Line 34. What does “complex” mean here? Either specify or delete.
Line 67. I believe “natural of” should be “native to.”
Line 69. Change to “mid-1990s.”
Line 69-70. Delete sentence beginning “Therefore.” It’s redundant.
Line 71. Change “for” to “to.”
Line 80. Again, avoid statements of primacy (“first time”). Maybe there is another paper in press right now. Or, maybe somebody did a study but it is published in an obscure venue. Suggestion: change “for the first time” to “new data on.”
Line 108. Change “eppendorf tubes” to “microcentrifuge tubes” and give the manufacturer and location. “Eppendorf” is the name of a specific company.
Line 145. Please provide all primer sequences.
Line 158. Change “to further evolutionary contextualization” to “for subsequent evolutionary analyses.”
Line 169. Change “golden lions tree” to something clearer.
Table 1. Change title to “Demographic and morphometric data…”
Lines 276-278. Delete this sentence because it is already in the Methods.
Lines 335-336. Again, avoid statements of primacy.
Lines 363-364. Including 95% confidence intervals in the results would help determine whether prevalence values fall into categories (non-overlapping confidence intervals).
Line 386. Typo: SUch.
Line 393. I believe the authors mean “massively parallel sequencing technologies.”
Line 395 and 426 and 430 and 432 and 433 (and maybe elsewhere). “Monophyletic clade” is redundant. Just say “clade.”
Lines 399-408. These are interesting ideas!
Line 411. Change “discarded” to “discounted.”
Line 444. Again, avoid statements of primacy.
Line 452. “zoonotic” means from animals to humans.
Lines 454-456. The two concluding sentences are not clear, and they are results rather than conclusions.
Response 5: We appreciate the suggestions and compliments about lines 399-408. All suggested corrections were carried out.
Point 6: Line 92. What does “allocated” mean here? I think another word might be clearer.
Response 6: The animals captured were free-living and are temporarily kept in captivity until they are transported to their natural habitat, in another state (Bahia). We substituted allocated for placed.
Point 7: Lines 174-184. How were these calibration dates chosen?
Response 7: The calibration dates were chosen considering the host evolutionary history (Kuderna et al., 2024), in points of our SFV phylogeny which mirrors the Primates major splits found by Kuderna (doi:10.1126/science.abn7829). We choose to use only the major points within the SFV Catarrhine splits, using the maximum number of calibrations and preferring calibrations closer to the root to an improvement in estimates of divergence times, following recommendations within Duchêne et al., 2014 (doi:10.1016/j.ympev.2014.05.032). To improve clarity, this part of the manuscript of the calibration node chosen was adjusted in the text (pages 5-6, lines 211-227).
Point 8: Line 206. 83.1 millimeters is a length, not a weight. Length measurements were not described in the methods, so it is unclear what this length refers to. Knee-heel distance as in Table 1, maybe? Please add details to methods and clarify.
Response 8: Thanks for the observation. The length was related to knee-heel distance, but as the same data was not available for L. rosalia and the information is not used in our analysis, we decided to remove them.
Point 9: Line 205. How was body condition assessed? If this statement is based on a formula including weight and length, please describe in the methods and add body condition score data to Table 1.
Response 9: Our veterinary collaborators basically measured the weight and knee-heel length of the animals and they observed whether they found any clinical signs of attention, such as: respiratory conditions (such as cold), diarrhea, alopecia and appropriate weight.
Point 10: Table 1. What does “Median collection per site” mean? Also, why are so many values for L. chrysomelas not available? This should be explained, because it suggests a problem with the field data (see general comments, above).
Response 10: We calculated the sampling average, considering the collection points. If this data is unnecessary or misleading, we can remove it. The sexual maturity of L. chrysomelas was added in the table (lines 253-254). Unfortunately, we don’t have access to the GPS coordinates, as explained above.
Point 11: Lines 259-260. Does the first sentence of section 3.3 mean that all samples yielded data (i.e. that no samples failed)? It’s not clear what is meant by “was possible to measure.”
Response 11: We agree that the sentence was poorly written, we rewrote it to make it clearer (Page 13, lines 358-360).
Point 12: Lines 282-284. This is a very low rate of sequencing success. The authors should carefully explain why sequencing failed for 49/50 samples. The authors should also explain why only 213 bp of sequence was obtained from a 404 bp fragment of integrase. As written, this section makes the reader suspicious that the samples that failed to sequence were false positives, perhaps due to nonspecific amplification during the qPCR. Also, failure to sequence the full amplicon (414 bp should be easy to sequence) makes t-he reader wonder about sequence quality and sequencing error. The authors should prevent convincing evidence that these issues were not the case.
Response 12: In fact, our sequencing has a low rate of success and occurs because we don’t have available sequences of genomes of SFV infecting L. rosalia or L. chrysomelas, which makes it difficult to design primers for SFV infecting this genus. For this specific PCR, only 4 of 50 positive samples were successfully amplified and forwarded for Sanger sequencing. From those, only one sample was properly sequenced, with only the reverse primer being functional. The primers used were constructed based on all complete sequences of SFV infecting American primates. That could result in primers that still catch the pol region of SFV from Leontopithecus, but it isn’t specific enough. For the qPCR, the primers amplify a smaller fragment, facilitating its amplification and diagnosis. The qPCR diagnose is comparable with serological tests, with a sensitivity that can reach 90 - 100% for 20 copies (Muniz et al., 2017, doi:10.1371/journal.pone.0184251).In our previous works (Miranda et al., 2019, doi: doi:10.3390/v11100931.; Muniz et al., 2013, doi: 10.1371/journal.pone.0067568.; Muniz et al., 2017, doi: doi:10.1371/journal.pone.0184251) it was demonstrated that the primers designed for amplifying a larger fragment of pol are not efficient for SFV infecting the Leontopithecus genus. In addition to that, the evolutionary history shown by our fragment is compatible with other data of literature (doi:10.1186/s12977-015-0214-0).
Point 13: Lines 333-335. There are problems with the phrase “and their diagnosis and monitoring can serve as a biomarker of zoonotic transmission of the virus between different callitrichid species.” First, “zoonotic” means from animals to humans. Also, how are diagnosis and monitoring a biomarker? There are also no references supporting this statement. I suggest deleting it or re-writing it to be clearer and supported by references.
Response 13: We changed “zoonotic” to “cross-species”. We decided to remove the sentence about the biomarker, because it is misleading.
Point 14: Lines 340-349. These are very good thoughts. But, without a multivariate statistical analysis, it’s difficult to know if the differences between the species are “real” or if they are explained by morphometric and demographic differences between the sampled animals of the two species.
Response 14: Thank you for your suggestion. Taking this into consideration, as explained above, we decided to do a multivariate statistical analysis of our data. We discovered that the difference in viral loads are not explained by any of our variables. Between lines 461 and 468 we discuss the results from the multivariate analysis.
Point 15: Lines 360-362. 42% is much higher than 29% or 22%, so why do the authors conclude that there’s no difference? The prevalence seems higher in adults.
Response 15: Our statistical analysis didn’t show any significance between these prevalences, neither in univariate nor in multivariate analysis.
Round 2
Reviewer 2 Report
Comments and Suggestions for Authors
Please see attachment.

Author Response
Girari et al., revised manuscript.
The authors adequately answered most points I raised. However, a careful proof-reading is necessary for both corrected and original texts. Effort to write shorter sentences have been done, but some sections are difficult to read because they are long or carry contradiction. Two examples:
Lines 122-125 : complex sentence.
Line 259-260 : the sentence states that geospatial coordinates are available, but these are not
available for L. chrysomela in the table.
Response: Thank you for your careful review of our manuscript and for your thoughtful comments and suggestions. We rewrote the sentence to make it shorter (lines 115-120). We apologize for not pointing out in the text that geographic coordinates were only available for L. rosalia. The text has been corrected (lines 245-246). We have thoroughly revised the English language of the manuscript, and we hope this will improve the clarity and comprehension of the text.
Point 1: Among changes introduced in the revised manuscript, two carry errors.
Statistics
First, the authors have added statistical analyses that carry several errors. These should be checked
by an expert in statistics. I guess that these novel analyses have been recommended by the other(s)
reviewer(s). However, I do not find them relevant.
Table 3 presents a binary logistic regression. The variable weight is related to species and age. Thus, its inclusion in the model does not fulfill one assumption for correct usage of the regression – i.e., no collinearity of predictor variables. Accordingly, stating that SFV prevalence is different according to weight is not biologically sound (Line 463-434).
As no multivariate analysis is performed, usage of binary logistic regression is superfluous and simpler statistical analysis (Fisher’s exact test) would be sufficient.
Response 1: We appreciate your thoughtful comments and concerns regarding the statistical analyses. We have carefully considered your feedback and contacted an expert in statistics that have checked the validity of the statistical methods used in this analysis. Specifically, we applied linear mixed-effect (LME) models to account for potential pseudoreplication by including species as a random effect when testing the predictors of proviral load. We chose to apply linear models because they allow us to include multiple predictors (age, sex, body weight) and provide more nuanced insights than Fisher’s exact test, especially for continuous and categorical variables. LME models are particularly suited for data with repeated measures or nested structures, which is why we deemed it appropriate for our dataset. To ensure the robustness of our model, we performed a series of diagnostic checks, including assessments of residual homoscedasticity, normality of the data distribution, and variance inflation factors (VIF). The results of these diagnostics were consistent with the assumptions of linear mixed-effects models: the data followed a normal distribution, residuals were homoscedastic, and the VIFs were consistently near or below 1, indicating no problematic multicollinearity. This is explained in Page 6, lines 256 - 264:
"We applied linear mixed-effect models to test whether morphological variables (age, sex and body weight) correlated with proviral load, including species as a random effect to control for pseudoreplication. The model was performed using the “lmer” function in the “lme4” package in R. To evaluate the validity of the model, we performed a series of diagnostic tests, including assessments of residual homoscedasticity, normality of the data distribution, and variance inflation factors (VIF), which were consistently near or below 1. The diagnostic results confirmed that the data followed a normal distribution, and no influential observations were detected".
Regarding the use of simpler analytical tests (such as Fisher's exact test), we indeed use them in the first part of our work (i.e., prevalence of SFV). This was done purposefully because our goal was simply to compare percentages among different groups.
Point 2: Figure 4 does not present the multivariate regression analysis: univariate analysis is absent and a single independent variable is presented while four variables were included in the model. The inclusion of weight as variable is inadequate as explained above.
Response 2: Indeed - this figure has been removed from the manuscript.
The following three mistakes illustrate that statistical analysis is misunderstood by the authors.
Point 3: Line 464-465 : the statement carries a contradiction.
Response 3: We revised the whole set of results to include the new analysis and avoid any misunderstanding and contradictions.
Point 4: Line 366 : according to the sentence, viral load was a dependent variable not an independent one.
Response 4: Indeed, viral load is the response variable in our linear mixed effect model, as explained in the reply above. We make sure that inaccurate sentences as this one has been removed or entirely revised.
Point 5: Line 369 – 371 : the results of the univariate analysis are given after those of multivariate analysis, which is awkward.
Response 5: In the updated version of the manuscript, multivariate analysis no longer exists.
Point 6: Viral load
Secondly, authors have modified the values of viral load in the text which are now around 4.4 and 4.5 log copies/mL (lines 358-360). These values correspond to figure 4. No changes were made in figure 3 and in other part of the text that present much lower values. The novel values no more support the conclusions.
Response 6: We feel sorry for the confusion. In some parts of the text the values were in scientific notation, but another reviewer suggested changing all viral loads to log10. The lines 335-338 show the mean viral loads. As the maximal values are high, the mean viral loads are higher too, but some individuals had low viral loads (1.29log and 1.43log being the lowest values). The values itself didn’t change, they were only converted to log.
Point 7: Line 30. Replace 0.0836 by 84 thousands
Response 7: We appreciate your consideration. As another reviewer suggested changing all datation values to Millions of Years Ago (Mya), to standardize dating analysis in the text, we decided to accept his/her suggestion.
Reviewer 3 Report
Comments and Suggestions for Authors
The authors have done an admirable job responding to reviewer critiques. The revised manuscript is highly modified from the original version and is much improved. I have just a few minor remaining suggestions:
Response 1, Response 12, and lines 390-396. Despite the authors’ explanation, it’s still not clear why sequencing failed for 3 of the 4 PCR-positive samples, and why only a 213 by sequence was recovered from the 404 bp amplicon. Contrary to what the authors state, this cannot be due to primer design and variability of SFV. If amplicons were generated, then the primers used in the PCR were present on the ends of those amplicons, and those primers should work fine for Sanger sequencing in all four cases. Also, 404 bp is not very long, so the internal amplification primers should have generated sequences spanning the entire amplicon. I’m guessing that the sequences were “messy,” which sometimes occurs, and that the authors removed messy segments based on the chromatograms but did not attempt to re-sequence in order to improve the quality of the sequences. There are many ways to troubleshoot this issue in the lab. If the authors can’t do additional lab work, they really should include a complete and honest explanation of why the issues above occurred.
Response 10 and Table 2. “Average collection per site” is still not clear. Is this the average number of monkeys sampled per site? Please add a footnote to the table to make clear what this means. Otherwise, it will confuse readers. Do keep the data in the paper, though.
Line 466. It’s not necessary to repeat P values in the Discussion.
Author Response
The authors have done an admirable job responding to reviewer critiques. The revised manuscript is highly modified from the original version and is much improved. I have just a few minor remaining suggestions.
Response: Thank you for your thoughtful comments.
Point 1: Response 1, Response 12, and lines 390-396. Despite the authors’ explanation, it’s still not clear why sequencing failed for 3 of the 4 PCR-positive samples, and why only a 213 by sequence was recovered from the 404 bp amplicon. Contrary to what the authors state, this cannot be due to primer design and variability of SFV. If amplicons were generated, then the primers used in the PCR were present on the ends of those amplicons, and those primers should work fine for Sanger sequencing in all four cases. Also, 404 bp is not very long, so the internal amplification primers should have generated sequences spanning the entire amplicon. I’m guessing that the sequences were “messy,” which sometimes occurs, and that the authors removed messy segments based on the chromatograms but did not attempt to re-sequence in order to improve the quality of the sequences. There are many ways to troubleshoot this issue in the lab. If the authors can’t do additional lab work, they really should include a complete and honest explanation of why the issues above occurred.
Response 1: Thank you for your comments regarding this issue. The total concentration of genomic DNA in the samples that amplified by PCR was low, ranging from 5.9 to 7.5 ng/μL. In the only successfully sequenced fragment, only the forward primer worked, generating a fragment of just 213 bp, as the remaining sequence presented several ‘noisy’ regions. Sanger sequencing was attempted three more times without further success. Additionally, we have faced some difficulties amplifying conserved genes of SFV infecting American primates, as well as their genomes. In light of these challenges, we are currently testing protocols for obtaining these genomes using high-throughput sequencing technologies.
Point 2: Response 10 and Table 2. “Average collection per site” is still not clear. Is this the average number of monkeys sampled per site? Please add a footnote to the table to make clear what this means. Otherwise, it will confuse readers. Do keep the data in the paper, though.
Response 2: We feel sorry for not being clear. We are referring to the average number of primates sampled per site. We added this information on a footnote of the table.